# Functional diversity of sharks and rays is highly vulnerable and supported by unique species and locations worldwide

Catalina Pimiento [1,2,3] ✉, Camille Albouy[4,5], Daniele Silvestro [6,7,8,9], Théophile L. Mouton [10,11], Laure Velez[10], David Mouillot [10], Aaron B. Judah [12], John N. Griffin [2,13] & Fabien Leprieur[10,13]

Elasmobranchs (sharks, rays and skates) are among the most threatened marine vertebrates, yet their global functional diversity remains largely unknown. Here, we use a trait dataset of >1000 species to assess elasmobranch functional diversity and compare it against other previously studied biodiversity facets (taxonomic and phylogenetic), to identify species- and spatial-conservation priorities. We show that threatened species encompass the full extent of functional space and disproportionately include functionally distinct species. Applying the conservation metric FUSE (Functionally Unique, Specialised, and Endangered) reveals that most top-ranking species differ from the top Evolutionarily Distinct and Globally Endangered (EDGE) list. Spatial analyses further show that elasmobranch functional richness is concentrated along continental shelves and around oceanic islands, with 18 distinguishable hotspots. These hotspots only marginally overlap with those of other biodiversity facets, reflecting a distinct spatial fingerprint of functional diversity. Elasmobranch biodiversity facets converge with fishing pressure along the coast of China, which emerges as a critical frontier in conservation. Meanwhile, several components of elasmobranch functional diversity fall in high seas and/ or outside the global network of marine protected areas. Overall, our results highlight acute vulnerability of the world's elasmobranchs' functional diversity and reveal global priorities for elasmobranch functional biodiversity previously overlooked.

Elasmobranchs (sharks, rays, and skates) are highly diverse, widely distributed around the world, and play many key roles in ecosystems[1]. Importantly, compared with other marine vertebrates, elasmobranchs are highly evolutionary distinct[2] and are greatly threatened by human activities, especially by overfishing[3,4]. Global conservation efforts are strongly resource- and space-constrained such that spatial (e.g., placement of highly protected marine areas over hotspots) or species-level prioritisation may ensure the greatest 'bang-for-buck' returns[5–7]. Yet,

while early prioritisation approaches focused on the taxonomic (species) level, a multi-faceted perspective to biodiversity evaluation and conservation prioritisation is crucial because evolutionary lability of functional traits may decouple taxonomic, phylogenetic, and functional diversity components[8]. Nevertheless, for elasmobranchs, global diversity and prioritisation studies have focused mainly on the taxonomic and evolutionary component[2,9], leaving the potentially independent−and ecologically relevant−functional component somewhat unexplored.

Functional diversity describes variation in species traits, and is a fundamental facet of biodiversity[10] known to be related to the capacity of organisms and communities to reliably supply functions and services including nutrient recycling and food provision[11]. Modern functional diversity analyses integrate species' attributes according to multiple ecological and life-history traits (e.g., body size, diet, habitat, etc.), providing a rich picture of species' functional differences[12,13]. However, species do not all contribute equally to functional diversity, with some holding traits highly dissimilar to others (functionally unique) and/or with extreme values (functionally specialised)[14]. The extinction of such species leaves large portions of trait space (and ecological roles) unoccupied and/or reduces the total range of traits present within the system[12,13], constraining the breadth of resources used and−potentially−services supplied[15]. Analogously to evolutionary measures[16], flagging species contributing inordinately to functional diversity has the potential to inform conservation prioritisation[17,18]. These functional trait-based priorities can be further refined by simultaneously considering the species' level of endangerment, as given by its extinction risk status as determined by the Red List of the International Union for the Conservation of Nature (IUCN)[18], or through indicators such as range size[17]. Such differences in the functional position or contribution of species may be further reflected at higher or coarser taxonomic scales (e.g., at the order level). Indeed, the ecological and evolutionary differences among elasmobranch orders has long been recognised[2,19,20].

Previous studies have shown that amongst the marine megafauna ( >45 kg), elasmobranchs are projected to suffer the largest losses of functional diversity if current trajectories are maintained[18]. Spatial analyses based on ecomorphotypes have further revealed global hotspots of shark (Selachii) functional diversity in Japan, Taiwan, the East and West coasts of Australia, Southeast Africa, Southeast Brazil, and Southeast USA[21]. Similarly, it has been shown that elasmobranch's hotspots for functional rarity (i.e., rare traits combined with geographical restrictiveness)[17] are distributed in the east coast of Asia, the southwest coast of Australia, south and west coasts of Africa, Central America, and the UK[22]. The evolutionary distinctiveness among elasmobranchs has also been investigated, suggesting a concentration of both species richness and evolutionary distinctiveness in tropical and subtropical coastal waters centred on four main areas: (1) Australia and the Indo-West Pacific biodiversity triangle; (2) Japan, China, and Taiwan; (3) southwest Indian Ocean; and (4) western Africa[2]. Furthermore, high congruence between elasmobranch evolutionary distinctiveness and endemism has been found in the (5) Gulf of California; (6) Gulf of Mexico, (7) Ecuador; (8) Uruguay; and (9) southern Brazil[9]. All of these areas have been suggested as regions of conservation priority[2,9]. In addition, it has been shown that evolutionary distinct species are concentrated mainly in two elasmobranch orders (Lamniformes and Hexanchiformes, or mackerel and cow sharks, respectively), with the most evolutionary distinct species being rays (i.e., the striped panray [*Zanobatus schoenleinii*], coffin ray [*Hypnos monopterygius*] and sixgill stingray [*Hexatrygon bickelli*])[9].

Despite recent advances, previous studies on elasmobranch functional diversity have exclusively included the largest species as part of the megafauna[18], have relied on coarse morphological groupings as proxies of functional traits while focusing on sharks only[21] (i.e., representing half of the total clade's diversity and excluding rays and skates, which are the most threatened group amongst chondrichthyans[3]), or focused only on one aspect of functional diversity (namely functional rarity)[22] or region of the world[23]. Thus, a global-scale evaluation of the multiple components of functional diversity (e.g., functional richness, uniqueness, and specialisation) of the elasmobranch clade is still lacking. Further, whether or not certain orders have greater functional diversity than others (and thus may warrant enhanced research and conservation attention) remains unexplored. Similarly, we do not know whether spatial patterns and

species-level priorities considering evolutionary disctinctivness[2] match those of functional diversity. The decoupling of phylogenetic and functional diversity has been reported for sharks, suggesting that closely-related species do not necessarily share the same ecological functions[24]. If this holds in the whole elasmobranch tree of life, ecologically relevant traits would decouple biodiversity facets and call for a re-evaluation or reconciliation of biodiversity hotspots and species-level priorities. Whether these priorities overlap with functional diversity and how they relate to threats from overfishing remains unknown. Importantly, despite a previous study showing a mismatch between hotspots of elasmobranch functional rarity and Marine Protected Areas (MPAs)[22], the extent to which different facets of elasmobranch biodiversity (functional, phylogenetic, and taxonomic) are protected by the global marine protected area network is yet to be assessed. Indeed, functional and phylogenetic diversity are rarely considered in protected areas research[25].

Here, we assemble a trait data set of 1015 elasmobranch species (~90% of total number of species[26]) to assess their functional diversity (richness, uniqueness, and specialization). We first describe the structure of the functional space across clades and threat status. We then quantify the contributions of individual species to functional diversity and apply the FUSE (Functionally Unique, Specialised, and Endangered) conservation prioritization metric[18], to identify highly threatened species whose global extinction would result in the most significant functional losses. We further assess the spatial distribution of functional diversity in relation to other facets of biodiversity (i.e., taxonomic and phylogenetic) and human impacts (i.e., fishing) to identify hotspots and congruence zones. Finally, we evaluate the extent to which the different facets of elasmobranch biodiversity are protected by the current global marine protected area network. Our results reveal the novel species and spatial conservation priorities that emerge when incorporating the functional dimensions of elasmobranch biodiversity, some of which occur in the high seas, highlighting the need to protect international waters, as proposed through the UN High Seas Biodiversity treaty.

## Results and Discussion
### Elasmobranch functional diversity
We assembled a dataset of 1015 elasmobranch species and assigned seven functional traits as well as their global extinction risk as provided by IUCN Red List statuses[27] to each (see Methods; Fig. S1) using primary literature, guides, technical reports, FishBase (www.fishbase.org/) and the IUCN Red List of Threatened Species (www.iucnredlist.org/). Missing trait data and extinction risk (IUCN Red List status = Not Evaluated [NE] and Data Deficient [DD]) were inferred using multiple imputations (see Methods). We performed a Principal Coordinate Analysis to build a three-dimensional trait space for the global elasmobranch assemblage representing 75% of the total inertia (or total variance; Fig. S2; Table S1). The first axis of the trait space is strongly related to habitat (i.e., coastal or oceanic; Table S2), with coastal, mostly riverine rays occupying the lowest values (e.g., the giant shovelnose *Glaucostegus typus* and the giant freshwater stingray *Himantura polylepis*) and oceanic sharks (i.e., living in the open ocean) the highest (e.g., the longfin mako *Isurus paucus* and the largetooth cookiecutter shark *Isistius plutodus*; Fig. S2). The second axis is mostly correlated with vertical position in the water column (i.e., benthic, benthopelagic, or pelagic; Table S2), with benthic rays occupying the lowest values (e.g., the Quilon electric ray *Heteronarce prabhui* and the Brazilian blind electric ray *Benthobatis kreffti*), and pelagic sharks the highest (e.g., the white shark *Carcharodon carcharias* and the whale shark *Rhincodon typus*; Fig. S2). Finally, the third axis of the trait space is mostly related to diet, with planktivorous elasmobranchs occupying the highest values (e.g., the lesser devil ray *Mobula munkiana* and the basking shark *Cetorhinus maximus*) and those feeding on invertebrates, fish and/or high vertebrates the lowest (e.g., the brown

ray *Raja miraletus* or the white shark; Fig. S2, Table S2). Elasmobranch functional diversity is therefore largely represented by three orthogonal axes of variation capturing the habitats occupied (habitat use), vertical position (relating to movement behaviour), and diet (trophic interactions and impacts on the food chain).

Based on the global elasmobranch three-dimensional functional space, we quantified functional richness (FRic = % volume of the trait space occupied), functional uniqueness (FUn = species' mean distance to the closest five neighbours) and functional

specialization (FSp = species' distance to the centre of the space)[14] across clades and global extinction risk as provided by the IUCN Red List. We further computed per-species FUn and FSp scores, and combined them with global endangerment by applying the FUSE metric to identify threatened species of particular importance for functional diversity[18]. Despite sharks (Selachii) having a smaller number of species than rays and skates (Batoidea), they were found to span a larger extent of the three-dimensional trait-space (sharks: 471 species, FRic = 68.2%; batoids: 544 species, FRic = 59.9%;

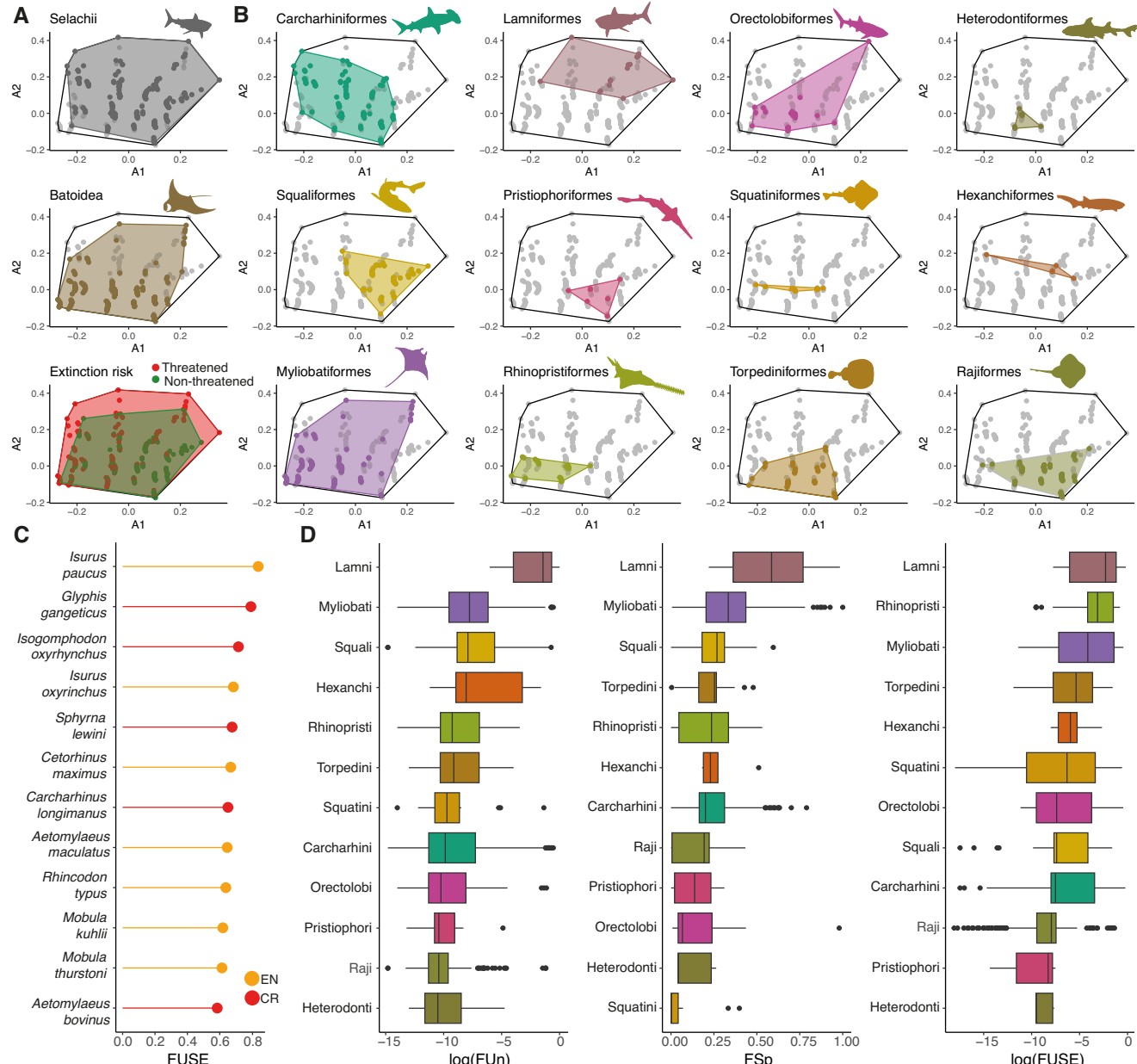

**Fig. 1 | Elasmobranch functional diversity. A** Functional space (first two axes) occupied by the elasmobranch clades (Selachii [sharks] and Batoidea [rays and skates]) and level of global extinction risk (red polygon = threatened species [IUCN status: VU, EN and CR]); green polygon = not threatened [IUCN status: LC and NT]). The animal shapes (downloaded from www.phylopic.org) illustrate the elasmobranch orders. They have a Public Domain license without copyright (http://creativecommons.org/licenses/by/3.0). **B** Functional space (first two axes) of the twelve elasmobranch orders. Dots inside the spaces in **A** and **B** represent individual species. **C** Top 12 FUSE species (see Fig. S6), orange colour denotes endangered species (IUCN = EN) and red critically endangered species (IUNC status = CR).

**D** Functional uniqueness (FUn), specialization (FSp) and FUSE scores of elasmobranch orders calculated per species and plotted in descending order from top to bottom based on the median of each clade. Orders are represented by the unique colours in **B** and **C**, and their names have been abbreviated in **C** by removing the *formes* suffix. FUn and FUSE were log-transformed to facilitate visualisation. Samples sizes (i.e., number of species) for each order are shown in Table S3. Median values are depicted by the black bar within the box. Whiskers depict the first and third quartiles of data. Outliers are defined as a black dot outside of the first and third quartiles.

Fig. 1A, S2; Table S3). Analyses across orders show that stingrays (Myliobatiformes) span the largest extent of global trait space (FRic = 53.7%; Fig. 1B). Indeed, this is the third most speciose order, with members ranging across a wide variety of sizes, habitats, and diets (Fig. S1). Mackerel sharks (Lamniformes) were found to be the most functionally unique and specialized order (Fig. 1D; Table S3-S4) as they exhibit extreme and uncommon trait-combinations across the elasmobranch functional space (Fig. 1B), including oceanic habits, disparate diets and feeding mechanisms (from filter-feeders to apex predators), and both ectothermic and mesothermic thermo-regulatory capabilities, the latter being unique for this order (Fig. S2). As such, lamniforms encompass the most unique and specialised species (top FUn and FSp), including the great white shark, the salmon shark (*Lamna ditropis*), the longfin and shortfin mako (*I. pacus* and *I. oxyrinchus*), the basking shark, the Porbeagle shark (*Lamna nasus*) and the megamouth shark (*Megachasma pelagios*; Fig. S6A, B). The high levels of lamniform functional uniqueness and specialisation contrast with their low species richness (only 15 living species; -2% of total diversity). This discrepancy can be explained by the fact that living lamniforms are the representatives of a once speciose clade that suffered high levels of extinction over the past 66 million years, while maintaining high levels of morphological ecological disparity[19,28].

Analyses of species' global extinction risk show that although 63% of species are not threatened (IUCN Red List status = Least Concern [LC] and Near Threatened [NT]; 637 species) they only occupy 48.2% of the functional space. In contrast, threatened species, which are the remaining 38% of the total richness (also see [31]) as inferred from multiple imputations (IUCN status = Vulnerable [VU], Endangered [EN] and Critically Endangered [CR]; 378 species) occupy almost the full extent (97.6%) of the functional space (Fig. 1A; Fig. S3; Table S5) and display significantly higher levels of functional specialisation and uniqueness (Wilcoxon-test $p < 0.005$; Fig. S4B). The level of threat is significantly associated with large body size, coastal habitats[3], freshwater incursions, and planktivorous diets (Fig. S5; Table S6). Guitarfishes, wedgefishes and sawfishes (Rhinopristiformes); angel sharks (Squantiformes); and mackerel sharks (Lamniformes) face the highest extinction risk amongst elasmobranchs (Fig. S4A). Elevated extinction risk among mackerel sharks, combined with their high functional uniqueness and specialization, makes them the order with the highest FUSE values followed by Rhinopristiformes and Myliobatiformes (Fig. 1D; Table S4). Nevertheless, the top 12 FUSE species span four orders (Lamniformes, Carcharhiniformes, Myliobatiformes and Orectolobiformes) and include the long fin mako (EN), the Ganges shark (*Glyphis gangeticus*, EN), the daggernose shark (*Isogomphodon oxyrhynchus*, EN), the shortfin mako (EN) and the scalloped hammerhead (*Sphyrna lewini*, CR) among others (Fig. 1C; Fig. S6D). FUSE scores were robust to the variation across trait imputations (Fig. 6F; see Methods). Our findings indicate that functionally unique and specialised species are more likely to be threatened and reveal the jeopardy that elasmobranch functional diversity faces. Conservation actions focusing on FUSE species are therefore essential to prevent the global loss of elasmobranch functional diversity.

We compared species' functional diversity and evolutionary distinctiveness (herein, ED) to explore the association between different facets of elasmobranch biodiversity. Unlike a recent analysis of elasmobranchs[2], when computing ED, we considered the extinction risk of close relatives by implementing the new ED2 metric[29,30]. We found a weak positive correlation between FUn and ED2 and FSp and ED2 (Spearman's rho = 0.2 and 0.1 respectively, $p < 0.05$; Fig. S7). Indeed, when comparing the top 20 FSp, FUn and ED2 species, we found no common species (Fig. S6). We further computed EDGE2 [i.e., Evolutionary Distinct and Globally Endangered][2,16,29] scores to compare top EDGE2 and FUSE species and found only two shared species amongst the top 20 rankings: the basking shark (*C. maximus*, the sixth

most FUSE and 17th most EDGE2; and the angel shark *Squatina squatina*, the 14th most FUSE and 20th most EDGE2; Fig. 1B, S6). Finally, we explored the distribution of top 20 EDGE2 and FUSE species in the functional space and found top EDGE2 species occupying low values along all three axes, unlike top FUSE species which are more widely distributed (Fig. S8). This segregation of top EDGE2 species in trait space suggests that species-based conservation priorities within the EDGE framework might not effectively capture the full breath of elasmobranch ecological functions. When taken together, our results suggest that elasmobranch ecological and evolutionary facets of elasmobranch diversity are largely decoupled.

## Biogeographic patterns of the different facets of elasmobranch diversity

Biodiversity can be evaluated at the *species-level*, where individual species receives a score indicative of their contribution to a particular biodiversity facet (as above), or at the *assemblage level*, where co-occurring species collectively receive a score for a particular biodiversity facet. While species level assessments motivate abatement of species-specific threats, spatial, assemblage-level measures can identify functionally diverse locations, and where functionally extreme and contrasting species co-occur. These insights can in turn allow the assessment of the extent to which these areas are captured by other biodiversity facets and whether they are adequately protected. Accordingly, we assessed the spatial patterns of elasmobranch functional diversity (i.e., FRic, FUn, FSp, and FUSE) and its associations with other biodiversity facets, specifically species richness (hereafter, SR), phylogenetic diversity (PD; minimum total length of all the branches required to span a given set of species on the phylogenetic tree, see Methods), phylogenetic uniqueness (PUn; mean phylogenetic distance of species to their closest neighbour, see Methods) and EDGE2. For this spatial-based analyses, we computed all metrics based on species co-occurring in each grid cell, with the FUn index being calculated using species' distance to their nearest neighbour, instead of their five closest, to ensure the inclusion of species-poor grid cells (see Methods). To perform our analyses we compiled the geographic range of all species arranged in 0.5° × 0.5° grid cell (~3000 km²) from the IUCN Red List of Threatened Species, computing each biodiversity metric per grid cell based on co-occurring species, and identifying and comparing hotspots, which are defined here as cells with the top 2.5% values (see Methods).

Elasmobranch FRic is highest in north-western Australia, with values globally concentrated along the global continental shelf, especially in the tropical belt and around oceanic islands (Fig. 2A). Our results are largely comparable to a previous study on sharks only (Selachii) based on eco-morphological richness, except that such work found the largest concentration of functional diversity in southern Australia[21]. This suggests that the different methodological approaches, and the inclusion of rays and skates (Batoids), may be driving the extreme FRic values of north-western Australia found here. We further found FUn to display an opposite spatial pattern, concentrating in high latitudes, especially in oceanic waters (i.e., open sea or off-shore) of southernmost South America, and showing the lowest values along the continental shelf (Fig. 2B). PUn spatial patterns largely mirror those of FUn, although the concentration of high PUn values in high latitudes and near the poles is more extensive (Fig. 2D). The contrasting spatial patterns of FRic and FUn may be partly explained by the diametrically opposed relationship of these metrics with species richness. Indeed, species richness is positively related with FRic (i.e., volume of trait space) but inversely related with FUn (Fig. S9), with assemblages having less species close to each other in the trait space - and hence less scope for functional redundancy- when they are species-poor[18]. However, we found a similar spatial pattern when computing FUn considering the mean distance of species to their nearest neighbour in the global pool of species (Fig. S10E), indicating that species poor-

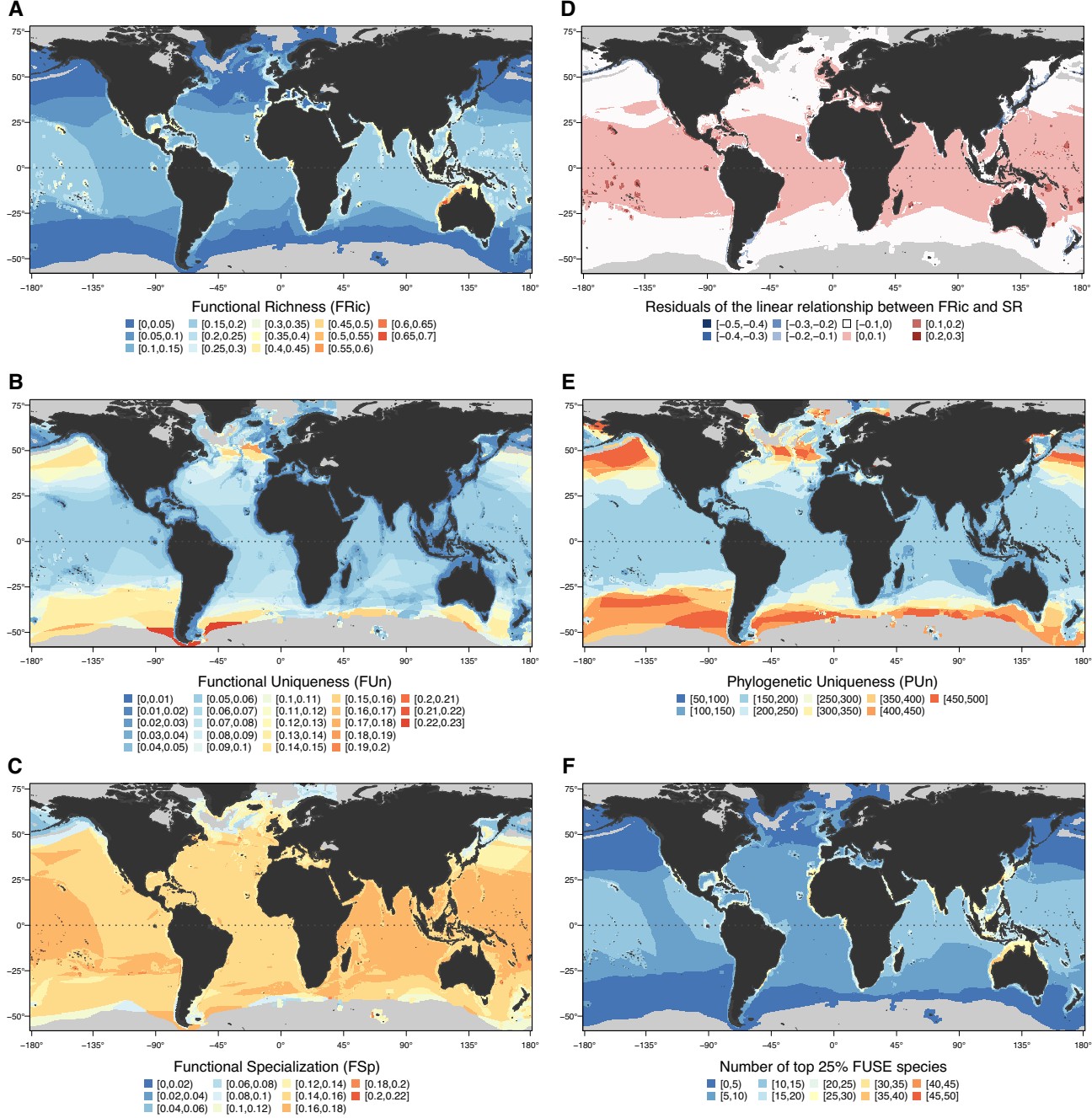

**Fig. 2 | Biogeographic patterns of elasmobranch diversity. A** Functional richness (FRic) per grid cell. **B** Functional uniqueness (FUn) per grid cell based on co-occurring species. **C** Functional specialisation (FSp) per grid cell. **D** Residuals of the relationship between FRic and Species Richness (SR) based on a linear regression model, with red in the colour gradient indicating greater FRic than expected, and blue indicating lower FRic values than expected (Fig. S10). **E** Phylogenetic uniqueness (PUn) per grid cell based on co-occurring species. **F** Distribution of the richness of the top 25% FUSE (Functionally Unique, Specialised and Endangered) species. All maps have been created using the R environment[51].

assemblages in high latitudes and oceanic waters are composed of species displaying unique ecological traits not only locally, but also globally. We further found FSp to display low spatial variation, with most grid cell values being near the global median. Despite the homogenous distribution of functional specialisation, high values can be identified along oceanic islands and the Indo-Pacific, and low values near the poles (Fig. 2C). Overall, our results indicate that the widest breadth of elasmobranch ecological traits is found along the tropical continental shelf, which is also where species-richness concentrates (Fig. S10A)[2,4,31]. However, species-poor assemblages in high-latitudes

and oceanic waters are composed of species displaying unique and specialised traits, and therefore possibly unique ecological functions, within local species assemblages (Fig. 2B, C, Fig. S10A)[18]. The high FUn found where few species co-occur indicates that the system has low functional redundancy, which implies that the loss of individual species is likely to leave large gaps in functional space[18]. It should be noted that, while we focus here on elasmobranchs, an alternative would be to eschew phylogeny and instead examine functional diversity across multiple clades that potentially contribute towards shared ecosystem functions (e.g., Pimiento et al.[18] and Waechter et al.[32]). The extent to

which species from other phylogenetic groups (e.g., large teleosts, marine mammals) alter global patterns of functional diversity, e.g., by providing redundancy, remains to be investigated.

Spatial patterns of species richness (Fig. S10A) fail to explain the distribution of elasmobranch functional diversity. Indeed, comparisons between different facets of elasmobranch biodiversity evidence high levels of spatial decoupling. For instance, the high levels of FRic in north-western Australia and oceanic islands are higher than expected given the SR values, as shown by the spatial distribution of residuals of the relationship between this index (Fig. 2D, S10C). This is also the case when comparing FRic and PD values (Fig. S10D). Similarly, FRic is lower than expected along the East China Sea, given the high SR and PD values of this region (Fig. 2D; S10C)[2,4,31]. The decoupling of FRic relative to SR and PD in Australia and oceanic islands suggests that species in these areas display a variety of ecological functions despite being closely related (i.e., functional overdispersion), while in the East China Sea, evolutionary distinct species span a smaller-than-expected extent of functional space (i.e., a functional deficit) suggesting either extreme trait conservatism or convergence[33]. The decoupling of functional richness from other facets of biodiversity is evident in the noisy, triangular relationship between these variables (relative to the linear and constrained SR – PD relationship; Fig. S9).

Despite such contrasting spatial patterns of functional diversity (FRic, FUn and FSp per grid cell; Fig. 2A–C) and the decoupling of biodiversity facets (Fig. 2D, S10D), the coasts of the Central Indo-Pacific region remain a key area for elasmobranch biodiversity where the highest concentration of both species' richness and threat (Figs. S10A, S11A) coincide[2,4,31]. As such, this is where the most (top 25%) functionally unique (Fig. S10B), specialised (Fig. S10C) and FUSE (Fig. 2F) species co-occur. Specifically, the highest richness of top 25% FUSE species occurs off Taiwan and the north and east of Australia (Fig. 2F).

Analysis identifying ocean cells with top 2.5% FRic values worldwide recognizes 23 hotspots: (1) Hawaii (USA); (2) The Line Islands; (3) French Polynesia; (4) the Pacific coast of Mexico; (5) the Galapagos Islands (Ecuador), (6) the Caribbean Sea and Western North Atlantic; (7) the coasts of Espíritu Santo, Santa Catarina and Rio Grande do Sul (Brazil); (8) The Canary Islands (Spain); (9) the south of Western Sahara; (10) the Strait of Gibraltar; (11) Ivory Coast; (12) Bioko (Equatorial Guinea) and São Tomé and Príncipe; (13) the southeast coast of Africa (from Somalia to South Africa); (14) northeast Madagascar; (15) Mauritius and Réunion; (16) the coast of Yemen and Oman; (17) the coasts of India, Bangladesh and Myanmar; (18) the Indian Ocean, specifically the coasts of Thailand, Cambodia, Vietnam, Indonesia, Malaysia; (19) Australia; (20) Luzon (The Philippines); (21) Hainan (China) and Taiwan; (22) Southern Japan; and (23) Vanuatu and Fiji (Fig. 3A). Notably, six of these hotspots occur around oceanic islands (i.e., hotspots 1–3, 5, 14 and 22). Congruence analyses found only a modest overlap between FRic hotspots and those previously identified for SR (36%; congruence test: $p < 0.001$; Fig. 3B) and PD (35%, congruence test: $p < 0.001$; Fig. S12A)[2,3,31]. As such, all hotspots in oceanic islands (except for some of the Canary Islands), as well as most of those in the Americas and Africa (hotspots 8-9 and some cells in 11–13, 14) are unique to FRic (Fig. 3A, B, Fig. S12B). In contrast, we found strong overlap between the SR and PD hotspots (84%, congruence test: $p < 0.001$). Notable areas of congruence for FRic, SR and PD include the Gulf of California, the coast of Louisiana and the Florida Keys (USA); Baja California Sur (Mexico); the Caribbean coast of Colombia (Bolivar and Magdalena); some parts of Rio Grande do Sul; most of the Canary Islands; the west coast of southern Africa; India, Bangladesh and Myanmar; and Luzon (Fig. 3B, S12B). These results indicate that although conservation efforts based on hotspots for species richness should be able to also protect phylogenetic diversity, they fail to capture priority areas for functional diversity, as most FRic hotspots, especially along oceanic islands, are unique.

Hotspots for FUn and FSp are found to be largely disassociated from territorial boundaries, with FUn hotspots extending along the highest latitudes where elasmobranchs occur, and across oceanic waters and high seas (Fig. 3D). FSp hotspots are distributed along oceanic islands of the Pacific Ocean and scattered around the Indian Ocean, with coastal hotspots in both the Atlantic and Pacific, specifically in Nova Scotia (Canada) and the Antofagasta region (Chile; Fig. S12A). When comparing FUn and PUn hotspots, we found only moderate spatial overlap (26% congruence test: $p < 0.001$), specifically in the north and south Atlantic and the south Pacific (Fig. 3C). These findings, combined with the weak correlation found between FUn and FSp vs. ED2 (Fig. S7), suggest that the functional and phylogenetic facets of elasmobranchs biodiversity are complementary, instead of redundant (Fig. 3C)[34].

We found strong spatial overlap (67%, congruence test: $p < 0.001$) between hotspots of FUSE and EDGE2. These hotspots were located exclusively in continental and island shelf areas. Overlapping hotspots were spread out mostly across the Indo-Pacific (northern Australia, much of the shelf from India to Southern Japan and areas of Indonesia and the Philippines), north-west and south-east Africa, Uruguay and southern-most Brazil and the Gulf of Mexico including the east coast of Florida (Fig. 3D). Such high level of congruence can be explained by: (1) the strong correlation between these metrics (and their individual components) when the number of species with the highest (top 25%) values are being considered (Fig. S13); (2) the higher species richness of these areas (Fig. S10A)[2,3], which increases the chances of finding highly scored FUSE and EDGE2 species; and (3) the fact that both the FUSE and EDGE2 scores are co-determined by global endangerment (i.e., GE as provided by their IUCN status), whereby highly scored species are necessarily also highly threatened[2,18,35]. Indeed, highly threatened species concentrate in the Indian Ocean and the Central Indo-Pacific (Fig. S11A)[2,3,31]. This suggests that although there is only moderate spatial overlap between FUn and PUn hotspots, the Indo-Pacific remains a key area for elasmobranchs, as it is not only where highly scored-FUSE species occur, but an area where FUSE and EDGE2 species greatly overlap (Fig. 3C). Nonetheless, we also found unique hotspots of FUSE, located in the Arabian Gulf (coasts of Pakistan and northern India, the central Red Sea, coasts of Oman and Yemen including Socotra Island, and coasts of Somalia), northern and southern Madagascar, open waters of the Canary Islands and areas of the Caribbean Sea (Fig. 3D). In contrast, unique hotspots of EDGE2 covered the southeast and south-west Australian shelve, much of the Indonesian island shelves, central and western South Africa, the Atlantic coast of the Iberian Peninsula and the Strait of Gibraltar (Fig. 3D).

Taken together, these results indicate that while the global continental margin harbours the largest number of hotspots for FRic, oceanic islands stand out as areas of great importance for elasmobranchs. Indeed, many unique hotspots for FRic and FSp, two metrics related with unique and/or extreme trait combinations[18], occur in these remote islands. While the continental shelf provides elasmobranchs with the greatest extent of shallow habitats which are key for marine biodiversity[36], oceanic islands provide them with: (1) high habitat heterogeneity provided by a wide range of depths concentrated in a relatively small area; and (2) geological dynamism as they tend to originate in tectonically active zones through hotspot volcanism, often resulting in chains of progressively older islands[37]. As such, elasmobranchs living in oceanic islands are likely adapted to an array of conditions, perhaps resulting in extreme trait combinations. Interestingly, oceanic islands have not been found to be hotspots for any previously studied facet of elasmobranch diversity such as ED or endemism (Figs. 2 and 3)[2]. Seeking to understand the mechanisms driving the high levels of elasmobranch functional diversity in oceanic islands may therefore represent a new frontier of elasmobranch research.

While our global analysis represents an important step towards prioritizing conservation actions in the regions where threatened

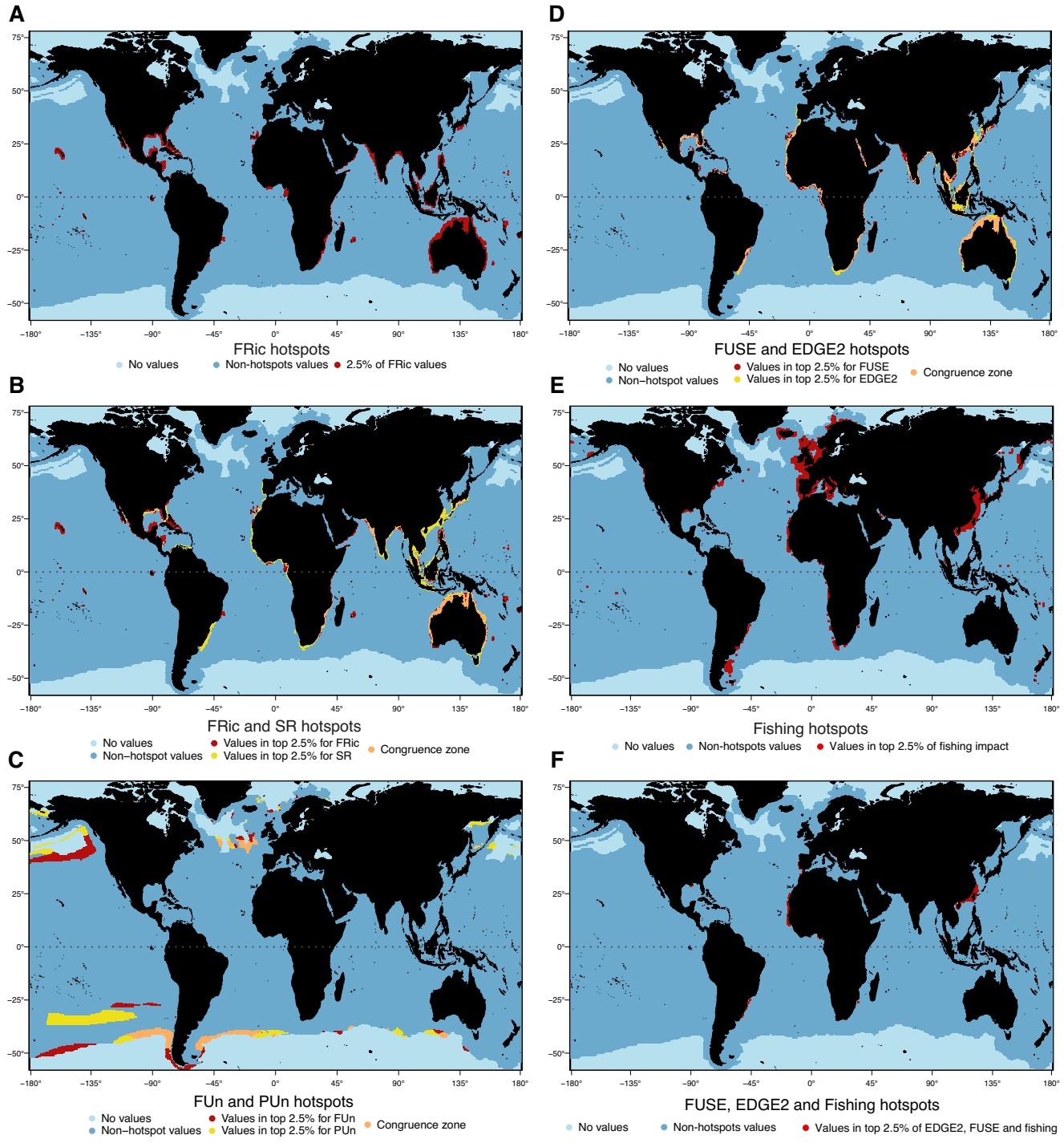

**Fig. 3 | Hotspots (top 2.5% cells) of elasmobranch biodiversity, fishing impacts, and their overlaps. A** Hotspots of functional richness (FRic). **B** Overlap (congruence zone) and non-overlap between FRic and species richness (SR) hotspots (orange = congruence; red = hotspots for FRic only; yellow = hotspots for SR only). **C** Overlap and non-overlap between Functional Uniqueness (FUn) and Phylogenetic Uniqueness (PUn) hotspots (orange = congruence; red = hotspots for FUn only; yellow = hotspots for PUn only. **D** Overlap between hotspots of richness for top 2.5% FUSE and EDGE2 species, respectively (orange = congruence; red = hotspots for FUSE only; yellow = hotspots for EDGE2 only). **E** Hotspots of fishing pressures (top 2.5% of cell values). **F** Overlap between fishing pressure hotspots and those for the richness of top 2.5% FUSE and EDGE2 species. All maps have been created using the R environment[51].

elasmobranch biodiversity is the greatest, it does not allow identifying biodiversity hotspots for each marine biogeographic realm[38]. Future studies could complement our findings by adopting a regional-based approach, where each region can be considered as independent pool of species. Coupled to finer distribution data at the regional scale (see Mouton et al.[39]) and to long-term abundance trends when available[40], this could ultimately help policy and decision makers, and

environmental managers, to set conservation actions towards threatened biodiversity at a regional scale (e.g., Mouillot et al.[8]).

## Human impacts and protection of elasmobranch functional diversity

While the above analyses identify global hotspots of multiple facets of elasmobranch biodiversity, a further crucial step is to identify the

exposure of these hotspots to anthropogenic threats. Accordingly, we used global databases of fishing pressure (from Global Fishing Watch's Automatic Identification System database of global industrial fishing effort[37], https://globalfishingwatch.org; Fig. S14) and Marine Protected Areas (MPAs; from the World Database on Protected Areas, https://www.protectedplanet.net/) and overlaid cumulative fishing pressures and gridded MPA locations with the hotspots of each biodiversity index (Fig. S15; see Methods).

In line with previous global analyses[41] we found that hotspots of fishing pressure (2.5% top cell values; see Methods) are mainly distributed along the coasts of China and Europe (both in the Atlantic and the Mediterranean). Smaller fishing pressure hotspots occur on the coasts of the US and Canada and along the Atlantic coast of southern South America (southern Brazil, Uruguay, and Argentina), west Africa and in the Sea of Okhotsk in Russia (Fig. 3E, S14). It is worth noting that the fishing pressure index used here does not cover the full extent of pressures in the world's oceans given that it only considers fishing fleets that use Automatic Identification System (AIS) technologies, which is geographically biased[42]. In addition, this index does not account for illegal and/or small-scale unreported fishing activities, which represent a major threat to sharks and rays[42,43]. As such, the areas identified here as fishing pressure hotspots represent a conservative estimate. Our analyses nevertheless identify the coasts of Louisiana (US), Southern Brazil, Cadiz (Spain), east India, Thailand's eastern Gulf, China and Brisbane (Australia), as areas where intensive fishing pressures overlap with hotspots of FRic, SR, and PD (Fig. S12C). Critically, the most substantial convergence between severe fishing, FUSE and EDGE2 extends along the coasts of China, Portugal and north-west Africa, and south Brazil (Fig. 3F, S12D). Overall, our results point towards global regions where elasmobranch biodiversity and those key species supporting it (top EDGE2 and FUSE) are inordinately threatened by fishing, with China being highlighted as an area of particular concern. Indeed, China's Exclusive Economic Zone has been identified as a priority area not only for biodiversity conservation, but also for carbon stocks and food provisioning[5]. Overall, our largely conservative results highlight the need to focus future research and monitoring in the Central Indo-Pacific, particularly, the South China Sea.

To evaluate the percentage of elasmobranch biodiversity hotspots overlapping with the global Marine Protected Area network, we considered "fully protected" MPAs only (Fig. 4A; see Methods, also see[39]). We found that the vast majority (89%) of MPA grid cells do not overlap with elasmobranch diversity hotspots (Figs. S16-S17). As such, 73–99% of elasmobranch biodiversity hotspot cells remain outside of the global MPA network, with notably 74% of FRic, and >90% of FUn, FSp, and PUn hotspot cells being unprotected (Fig. 4B, S18). Elasmobranch biodiversity hotspots are likely to be partially covered through MPAs around Australia[22] and through small MPAs in Southeast Asia and along the eastern US (Fig. 4A). However, the extensive FRic hotspots in the Arabian Sea and the smaller hotspot in the Philippines all fall outside of the existing MPA network (Figs. 3A and 4A). The lack of protection of FRic hotspots in the Arabian Sea and in South East Asia is particularly concerning considering the high landings of elasmobranchs and export of shark fins from these regions, as well as known high degrees of illegal, unreported and unregulated (IUU) fishing[3].

The poor protection of FUn and FSp hotspots is also of great concern as only 10% and 5%, respectively, fall within the MPA network (Fig. 4, S18). The lack of protection of these biodiversity facets probably reflects the concentration of FUn and FSp values in high seas where MPAs are lacking (Fig. 4A), and in pelagic temperate regions where the relatively few elasmobranch species hold unique functional roles and consequently low functional redundancy (Figs. 2–3)[22]. While these unprotected facets of biodiversity are not covered by AIS fishing hotspots (Fig. 3E), they are nevertheless exposed to high levels of longline fishing effort[41], which overlap with the habitat of oceanic

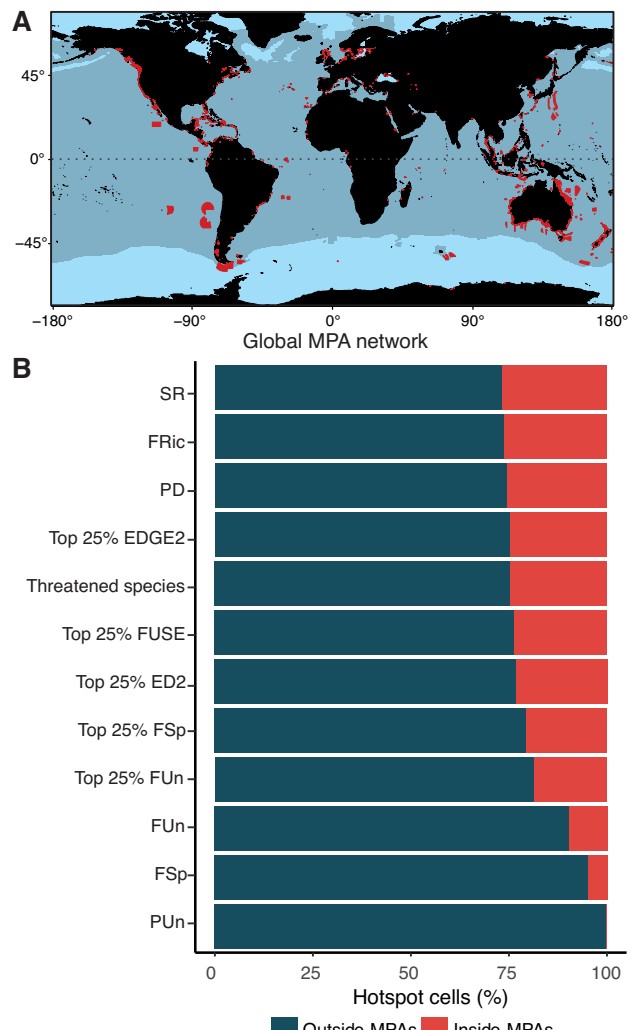

**Fig. 4 | Protection coverage of elasmobranch biodiversity. A** Marine Protected Areas (MPAs) status I to III ("fully protected"; see Methods). The map was created using the R environment[51]. **B** Percentage of hotspot cells for the different diversity metrics falling inside or outside MPAs.

sharks[44]. Indeed, oceanic sharks have declined as much as 71% since 1970 due to overfishing[45]. The need to protect biodiversity in international waters, as proposed through the UN High Seas Biodiversity treaty is evident from these results. Given the international push for expansion of protected areas through the UN Biodiversity COP 15 draft agreement to protect 30% of land and sea by 2030, there is clearly potential for redistribution, expansion, and optimisation of MPAs to maximise the protection of these highly threatened species. The identification of Important Shark and Ray Areas (ISRAs) is a promising tool for area-based conservation. These areas are discrete, three-dimensional portions of important habitats important for the survival of elasmobranch species[46]. Among the criteria proposed to delineate ISRAs, extinction risk and life-history are often used in conjunction, providing managers with areas that are critical for the survival of threatened elasmobranchs. The simultaneous use of hotspots of biodiversity and such expert-driven spatial planning tools offers new and modern avenues for the conservation of national and international waters, as part of a broader suite of necessary management measures that will be even more efficient if there is transboundary cooperation.

Biodiversity is well recognised to comprise taxonomic, evolutionary, and functional dimensions[47], yet the functional diversity and resulting conservation priorities of the oceans' most threatened

vertebrate group has to date remained unresolved. We here provided a comprehensive analysis of global elasmobranch functional diversity, underlining the unique information afforded by integrating multiple traits, and highlighting the emerging species and spatial conservation priorities. Our results show that, despite known correlations between traits and extinction risk[3], threatened species are inordinately varied, spanning the full extent of functional space. Among threatened species, we identified those most endangered *and* critically supporting the architecture of elasmobranch functional space, with the top five species (from four orders) comprising the long fin mako, the Ganges shark, the daggernose shark, the shortfin mako and the scalloped hammerhead. We further revealed multiple unique spatial hotspots for elasmobranch functional diversity and the species supporting it, including around oceanic islands and high seas. Alarmingly, we find that several aspects of elasmobranch functional diversity remain largely unprotected by the global MPA network, while several global regions hosting hotspots of all three biodiversity dimensions also face extreme threats from industrial fishing. Together, these results add a new dimension to our understanding of elasmobranch biodiversity and its conservation, revealing the extent to which it is decoupled from other better-studied aspects of biodiversity, and underlining the need to integrate functional elements into the protection of this—and other —highly threatened groups. Elasmobranchs have been a crucial component of marine ecosystems for hundreds of millions of years, surviving multiple mass extinctions and environmental changes[28]. The functioning of marine ecosystems as we know them depends on their outstanding diversity. Our characterization and mapping of functional diversity can help guide effective conservation action for sharks and rays to avoid a global extinction in the Anthropocene.

## Methods

### Elasmobranch data

We downloaded a list of elasmobranch species (sharks, rays, and skates excluding chimeras) from Fishbase (www.fishbase.org/; last accessed January 2019) using the R package *rfishbase*[48]. We checked all species names and contrasted them against Weigmann[26] to merge or eliminate names deemed to be synonyms. Further, we assign each species to a family and order based on Naylor et al.[49]. Our initial dataset included 1100 species. To each species, we assigned the following seven functional traits: maximum body size (total length or disk width), habitat (coastal, oceanic, or both), terrestriality (marine, brackish, or freshwater), vertical position (benthic, benthopelagic, or pelagic), diet (high vertebrates, fish, invertebrates, plankton or combinations of these), feeding mechanism (macropredators or filter feeders) and thermoregulation (ectothermic or mesothermic; Supplementary Information; Supplementary Data 1). Trait scores were assigned based mainly on primary literature (see Supplementary Methods) and to a lesser extent, on the species information provided by FishBase[48,50]. We based our selection of traits on Pimiento et al.[18] while tailoring the trait-assignments to elasmobranchs. The broad trait categorisations used allowed us to assign trait values to as many species as possible, enabling us to use a global-scale approach.

In addition to functional traits, we also assigned species threat status and geographical distribution, both gathered from the IUCN Red List of Threatened Species (www.iucnredlist.org/; last accessed October 2021). Finally, we downloaded the elasmobranch phylogeny ($n = 1192$ species of sharks, rays, and chimaeras) of Stein et al.[2], which was constructed using genetic data from 610 species. Species without DNA sequence were grafted according to a polytomy-resolver algorithm to generate a large distribution of fully resolved phylogenies ($n = 10,000$). We sampled 100 trees at random to account for phylogenetic uncertainty in the position of species without DNA sequence.

We performed all analyses described below using the R environment[51] based on a subset of 1015, which includes only those from which trait, spatial, and phylogenetic data was available, and excludes exclusively freshwater species (see Supplementary Information) and species that occur in grid-cells with less than four species (see below).

### Multiple imputations

Missing trait data and global extinction risk status (IUCN categories: Not Evaluated [NE] and Data Deficient [DD]) were imputed using the *missForest* R package, a non-parametric method based on random forests, here set to use 100 random trees [52]. Missing data included: habitat (7 spp.), terrestriality (or freshwater incursions; 1 spp.), diet (352 spp.) and size (1 sp); NE or DD IUCN status (170 spp.). We performed three types of imputations: (1) using known traits and taxonomic information to impute missing traits; (2) using geographical distribution and taxonomy to impute IUCN category; (3) using known traits, geographical distribution, and taxonomic information to impute all missing data. Given the small error (computed as number of incorrect/total number of imputed * 100) found for these three types (error for diet = 12% under imputation type 1 and 7% under type 3; error for IUCN category = 54% under type 2 and 49% under type 3); and the fact that the differences between imputed values using the different methods were minimal, we used the third type to perform all analyses. Family-level taxonomic information was included in the dataset by including one-hot-encoded columns attributing each species to their family. We repeated the imputations ten times and used the modal value across replicates as final trait prediction (Supplementary Data 2). Imputed values across the 10 iterations were highly consistent (Fig. S19).

### Species-level functional diversity metrics

To build the functional space and calculate the functional diversity metrics, we follow the methods used by Pimiento et al.[18] whereby we first created a trait dissimilarity matrix using a modified version of Gower's distance (using the function "dist.ktab" of the *ade4* package)[53] that allows the treatment of various types of variables (e.g., continuous, ordinal, nominal, multichoice nominal and binary) while giving equal weights to all traits considered (e.g., a trait with multiple categories does not have more importance than a trait with two categories)[54]. As we did not have a priori expectations regarding the relative importance of traits, we did not apply further weightings to traits. We then performed a Principal Coordinate Analyses (PCoA) to build a multidimensional Euclidean space. We selected the three first axes of the PCoA (Table S1), as they were deemed to provide the best compromise between a low number of axes and the least distortion of the original trait dissimilarity matrix[55].

On the basis of the multidimensional trait-space we first quantified the individual contributions of each species to functional diversity using the "fuse" function of the *mFD* package[56]. Specifically, we computed functional specialization (FSp; the distance of each species to the space centroid), functional uniqueness (FUn; the mean distance of each species to its nearest five neighbours); and FUSE (Functionally Unique, Specialized, and Endangered), which is the combination of FSp, FUn, and extinction risk[18] based on extinction probabilities in 100 years, as provided by species' IUCN categories[29,57]. We used the extinction probabilities as proposed in Mooers et al.[57] and not others recently proposed[58,59] to ensure a consistent comparison with EDGE2[29]. Given that we used modal values across trait imputations to perform our analyses (see above), we explored the uncertainty around FUSE values by computing this metric using each imputed dataset and calculating the mean and standard deviation. As anticipated given high consistency and correlation across imputed data, we found minimal variation around FUSE values (mean standard variation = 0.00241) suggesting that our ranking of top FUSE species is robust (Fig. S6F). Then, we computed functional richness (FRic; percentage of convex hull volume occupied in the functional space) per superorder (Selachii and Batoidea), order ($n = 12$) and global extinction risk status as

provided by species' IUCN categories (LC, NT, VU, EN CR; non-threatened [LC and NT] and threatened [VU, EN and CR]). Finally, we computed the mean FSp and the mean FUn[18] per order, superorder, and extinction risk status. This was done (1) based on mean distances in trait space and (2) based on species' individual contributions.

## Species-level phylogenetic diversity metrics

From Stein et al.[2], we randomly selected 100 trees among 10,000, to compute the mean evolutionary distinctiveness for each species. To do so, we used the ED2 metric proposed by Gumbs et al.[29], which is mathematically equivalent to the "Heightened Evolutionary Distinctiveness" (HED) metric[30] and considers the extinction risk of close relatives, thereby better reflecting the expected contribution of the species to phylogenetic diversity in the future[29,30]. We then combined ED2 with the IUCN status of each species to compute the EDGE2 metric [Evolutionarily Distinct and Globally Endangered], which have been recently proposed as an improvement of the EDGE protocol[29]. As for the FUSE metric, we used extinction probabilities in 100 years as provided by their IUCN status[57,58].

## Statistical analyses

To assess the association between PCoA loadings and trait values (Table S2) we used robust regression models[60]. We used this instead of a linear regression model because the residuals of the linear regression were not normally distributed (Shapiro-Wilk's test $p$-value < 0.05). To statistically compare threatened vs. non-threatened species according to their ED2, FUn and FSp values, we used a Wilcoxon-test (Fig. S4). To explore the association between species' extinction risk and their functional traits, we used binomial GLMs with extinction risk assigned to species based on their IUCN status (0 = non-threatened [LC and NT] and 1 = threatened [VU, NT, EN, and CR]; Fig. S6; Table S5). Finally, we measured the degree of association between per-species ED2 values vs. FUn and FSp values using a Spearman's rank correlation (Fig. S7). We used the above-mentioned non-parametric tests given that ED2, FUn, and FSp values were not normally distributed (Shapiro-Wilk's test $p$-value < 0.05).

## Spatial gradients of biodiversity

To perform assemblage-level analyses, we created the presence/absence matrix based on the geographic range of the studied species. We first quantified species richness (SR) by overlapping geographic ranges and counting how many species occur in each grid cell (0.5° × 0.5° grid-cells, ~3000-km²). We then calculated functional richness (FRic) per grid cell. We did this based on the 3D global functional space, whereby grid cells with less than four species were excluded[14,55]. We also computed two complementary functional diversity metrics reflecting the functional structure of local assemblages (i.e., species co-occurring in each cell), namely functional specialization (FSp) and functional uniqueness (FUn)[14,17]. FSp was calculated as the mean distance of each species to the centroid of the global functional space, hence reflecting the degree to which each assemblage contains species with extreme trait combinations. FUn was calculated as the mean distance of each species to their closest neighbour among the co-occurring species and therefore reflects the degree to which assemblages are composed of species with unique trait combinations locally[17]. Given the spatial scale dependency of metrics characterizing the functional uniqueness of species assemblages[17], we also computed, for comparison, the FUn index considering all the species in the global assemblage (i.e. the mean distance of each species to their closest neighbour within the global pool of species), reflecting the degree to which assemblages are composed of species with unique trait combinations globally[17]. Although our calculations of the FUn index per clade and species (see above) were based on the species' distance to their five nearest neighbours (i.e., NN = 5)[18], for our spatial analyses we calculated FUn

based on species' distance to their single most nearest neighbour (i.e., NN = 1) to ensure the inclusion of grid cells with only four species ($n = 5638$) and to avoid biased comparisons between grid cells with different number of species.

We also calculated mean phylogenetic diversity [PD; minimum total length of all the branches required to span a given set of species on the phylogenetic tree[61] per grid cell] based on the 100 random trees (see above) and phylogenetic uniqueness [PUn; mean phylogenetic distance of species to their closest neighbour among co-occurring species in each cell]. The PUn index is the phylogenetic counterpart of FUn, and it is also called the Mean Nearest Taxon Distance (MNTD) in the field of eco-phylogenetics[62]. This index quantifies the degree to which assemblages are composed of distantly related species towards the tips of the phylogeny. Finally, based on the species-level metrics (see above), we counted the number of species within the highest quartile (top 25%) of ED2, EDGE2, FSp, FUn, and FUSE scores per grid cell following Stein et al.[2].

## Fishing pressure

We built a fishing pressure index at 1° resolution from the Global Fishing Watch dataset (https://globalfishingwatch.org; last accessed December 2021) from 2012 to 2020.

This dataset consists of standardized apparent fishing effort based on transmissions broadcast using the vessel-tracking system (Automatic Identification System, AIS) originally designed for collision avoidance. Apparent fishing effort is calculated based on fishing hours of all fishing vessels detected in a given area. To compute our fishing effort index, we summed the hours spent at sea of vessels using fishing types known to potentially target elasmobranch species: pole and line; drifting longlines; tuna and other purse seines; trollers; other seines; set gillnets and longlines; and trawlers. It is worth noting that the fishing pressure index used here does not cover the full extent of pressures in the world's oceans given that it only considers regions where the AIS is in place[41]. In addition, this index does not account for illegal and small-scale unreported fishing activities, which represent a major threat for sharks and rays[42]. To perform spatial congruence analyses between fishing effort and the different biodiversity metrics, we applied an ordinary kriging technique, whereby a distance of 3° around each location was chosen for fishing effort interpolation. This distance constraint was utilized to minimize missing values in our fishing pressure index and to align the resolutions of the different datasets (biodiversity metrics and fishing effort).

## Spatial congruence analyses

To map the spatial congruence between all the elasmobranch biodiversity facets and fishing pressure, we quantified the spatial overlap between hotspots[8], using pairwise comparisons. This analysis allows identifying if two biodiversity facets present similar spatial repartition of high values, which is not straightforward with a correlation coefficient that only evaluates the degree of dependence between two quantitative variables. We defined hotspots as all grid cells with values in the upper 2.5% of the biodiversity facets following previous works[2,8,63,64], and 2.5%, 5%, and 10% of fishing pressure values. For example, for a pairwise comparison between SR and FRic, we calculated the observed number of overlaps, which corresponds to the number of cells which are recorded as a hotspot for both indices, expressed as percentage. Then, we evaluated the expected number of overlaps ($Oe$), corresponding to the independence between the hotspots of the two indices. This was calculated as follows:

$$Oe = N^i \times N^j / N^T \tag{1}$$

where $N^i$ is the number of hotspots for one biodiversity index (e.g., SR), $N^j$ the number of hotspots for the second index (e.g., FRic), and $N^T$ is the total number of grid cells. We then performed a randomization

procedure to assess whether the observed number of overlaps ($Oo$) was significantly different from that obtained by chance ($Oe$). Values contained in cells of one of the two variables considered were randomly permuted 999 times and the number of overlaps was estimated for each.

## Marine protected areas and overlap with biodiversity

The Marine Protected Area (MPA) database was downloaded from the World Database on Protected Areas (https://www.protectedplanet.net/en, last accessed January 2023). From this first set of MPAs, we retained only MPAs that passed the following sequential filtering criteria ($n = 3,376$). First, we adopted a conservative approach by selecting MPAs with IUCN categories I to III only (i.e., Ia, Ib, II, III), which correspond to three management and governance types[65], and are fully protected marine areas[66]. Although commercial or industrial fishing is not allowed in any MPA, we did not consider partially protected marine areas (i.e., categories IV, V, and VI) because fishing activities are allowed in these areas as long as they are managed and meet MPA objectives. Second, we deleted MPAs designated to protect species not considered in this study (e.g., birds or flora) by inspecting the 'Designation' field (DESIG_ENG) of the MPA shapefile (see Supplementary Information). We performed two sets of analyses. First, we overlayed gridded MPA locations with the hotspots (using 2.5%, 5%, and 10% thresholds) of each biodiversity index and quantified the percentage of overlap (i.e., the hotspots cells inside MPAs; Fig. S15A). Second, we measured the diversity represented within each protected grid cell, following Mouton et al.[39]. This approach is a synthesized and continuous assessment of the elasmobranch biodiversity contained in MPAs. It allows us to assess whether MPAs overlap areas of high biodiversity. For each biodiversity index, we extracted all grid cells overlapping MPAs, which we ranked from the least to the most diverse. We then plotted these ranked values of protected shark biodiversity against the cumulative percentage of protected area (Fig. S15B).

## Reporting summary

Further information on research design is available in the Nature Portfolio Reporting Summary linked to this article.

## Data availability

The data used in the study is included in Supplementary Data 1 and 2, and in the "output" files included with the R-codes (see below). The list of species used in this study was downloaded from FishBase (www.fishbase.org). Traits were assigned to species based mainly on primary literature (see Methods and Supplementary Information). Geographical distribution to create the maps (Figs. 2–4) was downloaded from the IUCN Red List of Threatened Species (www.iucnredlist.org). The elasmobranch phylogeny was accessed from Stein et al.[2]. The fishing pressure data shown in Fig. 3e, f was accessed from the Global Fishing Watch (https://globalfishingwatch.org). The Marine Protected Area data used in Fig. 4 was downloaded from the World Database on Protected Areas (https://www.protectedplanet.net/en).

## Code availability

The R codes used in the study have been deposited in GitHub, along with all input data and output files necessary to reproduce all analyses and figures: https://github.com/Pimiento-Research-Group/sharks-FD_biodiv_global.

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

## Acknowledgements

C.P. thanks L. Naidenov and D. Spitznagel for their help accessing some of the data and C. Jaramillo for providing valuable insights. This project received funding from the European Union's Horizon 2020 research and innovation programme under the Marie Skłodowska-Curie grant agreement (no. 663830) to C.P. C.P. is now funded by a PRIMA grant (no.185798) from the Swiss National Science Foundation. D.S. received funding from the Swiss National Science Foundation (PCEFP3_187012), the Swedish Research Council (V.R.: 2019-04739), and the Swedish Foundation for Strategic Environmental Research MISTRA within the framework of the research programme BIOPATH (F 2022/1448).

## Author contributions

C.P., J.N.G. and F.L. designed the research work. C.P., C.A., T.L.M., L.V., D.M. and A.B.J. gathered the data. A.B.J. curated the trait data. C.P., C.A., D.S., T.L.M., L.V. and F.L. performed the analyses. C.P. and J.N.G. led the writing with input from F.L. All authors contributed to the preparation of the manuscript.

## Competing interests

The authors declare no competing interests.

## Additional information

[1]Department of Paleontology, University of Zurich, Zurich, Switzerland. [2]Department of Biosciences, Swansea University, Swansea, UK. [3]Smithsonian Tropical Research Institute, Balboa, Panama. [4]Ecosystem and Landscape Evolution, Institute of Terrestrial Ecosystems, Department of Environmental Systems Science, ETH Zurich, Zurich, Switzerland. [5]Unit of Land Change Science, Swiss Federal Research Institute WSL, Birmensdorf, Switzerland. [6]Department of Biology, University of Fribourg, Fribourg, Switzerland. [7]Swiss Institute of Bioinformatics, Fribourg, Switzerland. [8]Department of Biological and Environmental Sciences, University of Gothenburg, Gothenburg, Sweden. [9]Gothenburg Global Biodiversity Centre, Department of Biological and Environmental Sciences, University of Gothenburg, Gothenburg, Sweden. [10]MARBEC, Univ Montpellier, CNRS, Ifremer, IRD, Montpellier, France. [11]International Union for Conservation of Nature Species Survival Commission Shark Specialist Group, P.O. Box 29588 Dubai, United Arab Emirates. [12]Department of Biology, Dalhousie University, Halifax, NS, Canada. [13]These authors contributed equally: John N. Griffin, Fabien Leprieur. ✉e-mail: catalina.pimientohernandez@pim.uzh.ch

