## [Peer Review File · Nature Communications]

Functional diversity of sharks and rays is highly vulnerable and supported by unique species and locations worldwideREVIEWER COMMENTS

Reviewer #1 (Remarks to the Author):

Thank you for the opportunity to review this very interesting paper. The paper is very well written, presented, and I enjoyed reading it. The paper seemingly fits the scope of the journal and promotes the application of function diversity in elasmobranch conservation prioritisation. Overall, I think this is a strong manuscript and, with a few minor adjustments, would be fit for publication. Below, I provide some comments and feedback that I hope the authors find useful.

Line 52 - Reference needed to support this statement/sentence.

Line 58 – Not entirely unexplored, especially given the co-authors publication history. I would recommend downplaying this statement a touch.

Line 95 – It is not clear how the two halves of this sentence are linked. Further, I am not sure that the reference given at the end is the most relevant to support these statements, given it focusses on spatial distributions.

Line 133 - Given the introduction of the UN High Seas Biodiversity Treaty, it would be interested to have a direct comparison between hotspots withing and outside EEZs.

Line 118 – The mismatch they refer to here should be made clearer.

Line 164 – Could the authors double check the FRic scores here. Visually, shark trait space looks larger than batoids in Figure 1.

Line 252 – Very little justification is given for the decision to focus on only the top 2.5% cell values, especially for fishing pressure. This seems to be somewhat selective and based on an arbitrary threshold. Why did the authors adopt this approach, rather than using them as continuous variables instead?

General point – There is a heavily reliance on Fish Base derived data to inform the trait database. However, some of the data in Fish Base is questionable and coverage is patchy. The authors do a good job in plugging gaps from the known literature but could have gone further with sourcing primary data to inform their trait database. For example, the recent study in the vertical distributions of elasmobranchs (see Andrzejaczek et al 2022). This is pertinent as the vertical position categories are too broad and more refinement/bins would better link to function. Admittedly, it may not be possible to entirely re-run the analyses. However, the limitations of taking a broad-brush approach should be addressed in the manuscript.

Andrzejaczek, et al. 2022. Diving into the vertical dimension of elasmobranch movement ecology. *Science Advances*, 8(33), p.eabo1754.

Line 501 – What was the justification for including only these variables and not others, such as swim speed, trophic position (instead of diet which, looking at Table S1, doesn't contain much variability), a measure of horizontal distribution, or thermal tolerance (more than ecto vs endo).

Line 548 – The authors note that all traits were weighted equally within the analysis. However, the implications of this assumption are not discussed within the manuscript. As with other comments here, more discussion on the limitations of their general approach would be welcome in the discussions.

Line 554 - There has recently been an update EDGE approach, referred to as EDGE2. Given the lack of congruence, it would be interesting to see if this changes much in the analyses. At the very least, the shortcomings of the old approach and the potential of this new metric should be discussed within the paper.<https://journals.plos.org/plosbiology/article?id=10.1371/journal.pbio.3001991>

Lines 558 – More information is needed to demonstrate that the statistical approaches are valid (i.e., normally distributed)

Line 583 – Could the authors provide more information on how they summed these different fishing types, many of which have very different units of effort associated with them (i.e., days fished, number of hooks, set time, etc).

Line 583 – Also, were data obtained at the same spatial resolution at the SR calculations? If not, what steps were taken to align these.

Lin 585 – This statement should be supported by a reference or more information. Whilst true to an extent, I think it is unfair to solely point the finger of blame to SE Asia. There are plenty of issues with the GFW dataset and the AIS data source. A more balanced and comprehensive discussion of those should be added here (or in the discussion).

Line 617 – I am not familiar with this ‘Designation Field’ but is there a risk that authors removed some MPAs that use such species as umbrella or flagship species to conserve biodiversity more generally? What does the overlap look like if these are included? Also, how many and what spatial coverage was removed by taking this step?

Figure 1C – What is determining the order of the y-axis here? Would better to order each by x-axis mean in A, then carry that forward in B, C. This could also free up some space to detail the species names.

The overall narrative of the paper is understandably focussed on elasmobranchs. However, if we are truly to shift to a functional based priority system for conservation action, then is it appropriate to even focus within a taxonomic group at all? Similar or same functions can be realised/provided by species in other, even less related branches of the tree of life (e.g. large teleosts or marine mammals). Thus, the argument could and should be made that a more holistic based approach should be adopted, rather than focussing on elasmobranchs. I would like to see some more discussion on this topic added to the manuscript.

Reviewer #2 (Remarks to the Author):

Dear editor, thank you for the opportunity to review the paper “Functional diversity of

sharks and rays is highly vulnerable and supported by unique species and locations worldwide.”. This is an interesting study, where the authors are putting together multiple facets of biodiversity (taxonomic, functional and phylogenetic diversity), not only at species level (calculating diversity indices for all species individually) but also at assemblage level (here the authors adopted grid cells as alpha diversity and the global pool of species as gamma diversity).

Indeed, the scientific community is constantly showing that global biodiversity conservation will benefit from considering multiple facets of biodiversity for improving conservation and global fisheries management. Another important aspect is to consider that biodiversity indices (taxonomic, functional, or phylogenetic diversity) are scale dependent, several papers showing this relationship are published. Here the authors did not consider this scale dependence. This study would benefit by dividing the global oceans in ecoregions and use those ecoregions as the gamma diversity. This is a major issue, undermining the study interpretations. Please see detailed comments below.

Lines 29 – 31. Suggestion: Maybe this can be rewritten as follows - "Conservation priorities are more efficiently applied at species and spatial levels (taxonomic diversity) and considering complementary biodiversity facets such as functional and phylogenetic diversity. Here, we address this challenge by using a trait dataset of > 1,000 species to assess the conservation status of the multiple facets of elasmobranchs biodiversity."

Lines 58 – 60. Provide reference.

Lines 88 – 92. Use reference number 2 at the end of this statement as well.

Lines 112 – 116. This is a strong statement based on only one reference looking at two diversity metrics. Ancestral state reconstruction or the fossil record can answer this question, as can clustering (which would inform ancestral states) or phylogenetic independent contrasts of correlations between traits. However, a mismatch between phylogenetic and functional diversity does not necessarily mean this at all.

Line 146. "Total inertia" - Is it possible to provide any clarification to the reader about what inertia represents in this context?

Line 149. "Vertical position in the water columns" - It needs to be clarified this is independent of environment, otherwise you would expect a deep-sea shark to be the species with the lowest value on this axis.

Lines 179 -183. This is a confusing conclusion, as they must have rediversified after the loss of marine reptiles if actually affected by the extinction, and because above the authors claimed there was no conservation of ecology over time in sharks, yet here are positing ecology was conserved for the last 66 million years. Also, the loss of other species would not lead to the group being the most disparate in terms of space occupation - instead the space would just be filled with more species but the convex hull unchanged (or the occupied space would have been even larger in the past). It might affect FUn only, but not Fsp, as the center would be unchanged.

Line 190. "Occupy almost the full extent (99%) of the functional space." - The functional space shown at the Figure 1 A represents the functional space occupied by all species. That is = 100% of the species included in this study (100% = 1015 species). However, the functional space shown for the IUCN panel (bottom left) does not seem to display the same functional space structure as the other panels (constructed using all species as the other ones). Here red is the functional space occupied by threatened species (631 species) and green is the non-threatened (283 species). We then have $631 + 283 = 914$ species, this means that this new functional space shown in this panel does not represent the 1015 species. Therefore, when the authors say that threatened species "occupy almost the full extent (99%) of the functional space" this new percentage given does not refer to the functional space for all species. In fact this new percentage refers to 914 species ($1015 - 914 = 101$ species less in comparison with the remaining panels). This may lead the reader to not accurate interpretation. Is there a reason why the authors are not using the same functional space structure in this IUCN panel as in the other ones?

Lines 233 – 239. It is not clear how the conclusion follows from this? The most endangered

are coastal species, but is that because there is insufficient data in the EDGE metric for rarely studied species like deep sea sharks, Hexanchiforms, or megamouths? How much of this is decoupling and how much of this is that the EDGE metric just isn't very good for sharks, or based on a bad phylogeny or taxonomy? In other words, is FUSE just a better metric for both ecological and evolutionary distinctiveness overall than the confounded and partially overlapped EDGE (which combines two unrelated metrics)? If so, then a mismatch cannot be taken to mean decoupling of anything.

Line 242. Maybe start this paragraph reminding the reader that there is two ways of measuring diversity - at species level (as you shown previously, in this case each species receive an index) and at assemblage level (as you are now proposing here in this section, in this case each assemblage receive an index).

Line 247. "FRic, FUn, FSp, and FUSE" Here the authors are not considering something important. Those metrics are scale dependent, here you are not comparing species from a defined ecological scale, such as Large Marine Ecosystems, Marine Ecoregions or other ecological classification. Instead, you are using the global scale as your gamma diversity and the grid cells as your alpha diversity. You are comparing a species from North Atlantic with a species from South Australia. This study would benefit by dividing the worlds oceans into specific ecological regions (biogeographical/environmental criteria) and then use each of those as your gamma diversity.

The suggestion here is:

- 1 - Divide the worlds ocean into ecoregions.
- 2 - Calculate the indices at species level and assemblage level for each of those ecoregions separately.

Line 268. "The low..". Is it low or high?

Line 309. "Spiritu Santo" – Espírito Santo.

Lines 475 – 478. I am not sure what this means, or why it is important. The main axes were habitat, water depth, and diet, but it seems like they use the metrics drawn from their space to make a lot of other inferences about how ecologically unique a species is. There is no

comparison of species in similar space or habitat along other traits, which would be required to make this claim, and there is no examination of close relatives.

Lines 487 – 489. Earlier (lines 179 - 183) the authors said that Lamniform diversity is the result of being nearly wiped out by the end-Cretaceous...and that there is no conservation of ecology in sharks. The same statement could be made about tetrapods or teleosts and it would be equally facile. Sharks were also freshwater before the Mesozoic...

Line 691. “global diversity” – Global diversity.

Reviewer #3 (Remarks to the Author):

The authors analyse the functional diversity of sharks and rays (~1,100 species) across the globe using multiple functional diversity metrics (functional richness, functional uniqueness, functional specialization), as well as comparing functional diversity to other biodiversity facets (species richness, phylogenetic diversity, evolutionary distinctiveness). They find that threatened species disproportionately contribute to functional diversity. They also identify multiple unique elasmobranch functional diversity spatial hotspots (not represented by other biodiversity facets), including around oceanic islands and high seas. Yet much of this elasmobranch diversity is unprotected through the global MPA network. Overall, I think this is an interesting study that covers a lot of ground in terms of biodiversity metrics and spatial prioritization. Even though there are a growing number of papers on this topic (Stein et al., 2018; Derrick et al., 2020; Lucifora et al., 2011; Pimiento et al., 2020), I think it is a useful addition to the literature. In particular, I think the authors have done a particularly good job of presenting multiple metrics of biodiversity and multiple functional diversity metrics for all elasmobranch species (previous papers generally don't cover all elasmobranch species, or, where they do, they generally focus on taxonomic and phylogenetic diversity, or single metrics of functional diversity). I also think that the manuscript is well articulated and the analyses generally well executed. Still, I have quite a few comments, which I have listed below (line number in brackets). To be clear, these are suggestions and I leave these up to the discretion of the authors who will know their study much more closely than I do.

Comments:

(30) 'assess elasmobranch functional diversity' – personally, I think it would be informative for the reader to know which metrics of functional diversity have been used, as functional diversity means different things to different people. Perhaps: "assess elasmobranch functional diversity (functional richness, functional uniqueness, and functional specialization)".

(30) 'compare it against other previously studied biodiversity facets' – again, it is unclear what other biodiversity facets were investigated from the abstract alone. I understand that space is limited in an abstract, but I think this information would be really useful for the reader to set the analysis in context.

(44) 'elevated functional vulnerability of the world's sharks and rays' – vulnerability is not mentioned in the main text. I also assume that the authors mean vulnerability in the general sense, rather than referring to more explicit implementations to functional vulnerability (e.g., Toussaint et al., 2016; Auber et al., 2022). I think it needs to be much clearer in the main text what is being referred to by the term functional vulnerability, or vulnerability should be dropped or reworded in the abstract.

(58) 'Nevertheless, for elasmobranchs, global diversity and prioritisation studies have focused mainly on the evolutionary component' – for me this needs referencing, what studies are you referring to?

(124) 'Here, we assemble a trait data set for over 1,000 elasmobranch species to assess their functional diversity.' – I think it would be good to include a percentage here for how many of the total species are included. E.g. "Here, we assemble a trait data set for over 1,000 elasmobranch species (X% of all elasmobranch species) to assess their functional diversity."

(126) 'We then quantify the contributions of individual species to functional diversity and apply the FUSE (Functionally Unique, Specialised, and Endangered) conservation prioritization metric^{14 128}, to identify highly threatened species whose extinction would result in the most significant functional losses.' – I would be included to change this to "... whose global extinction ...", just to be clear. As when considering local or regional extinctions the importance of species may be different (different context of extinction).

(142) 'assigned seven functional traits' – I think it would be useful to specify which traits were used here.

(159) 'FRic = % volume of the trait space occupied'. What does this volume refer to? How was this volume calculated? Volume is not mentioned in the Methods and it is unclear

exactly how this metric was calculated. As there are many ways to calculate the volume of a trait space available (e.g., hypervolumes, convex hull, kernels) I think more detail is needed here/in the Methods.

(169) 'with members ranging a wide variety of sizes, habitats and diets' – probably should be “with members ranging across a wide variety of sizes, habitats and diets”

(170) 'As such, stingrays have the most extreme trait-combinations (FSp) within their own functional space (Fig. 1A; Table S3).' – what do you mean by their 'own' functional space? Aren't all species compared across the same trait space, whereas this wording implies that each group had a separate trait space built? Perhaps, “As such, stingrays have the most extreme trait-combinations (FSp) across the elasmobranch functional space (Fig. 1A - Myliobatiformes; Table S3).” I also added the order name again just to make it clear what part of Fig 1A is being referred to here.

(199) 'Nevertheless, the top five FUSE species span five orders (Myliobatiformes, Rhinopristiformes, Lamniformes and Carcharhiniformes) and include the Chilean devil ray (*Mobula tarapacana*, EN), the broad-nose wedge-fish (*Rhynchobatus springeri*, CR), the long fin mako (EN), the lesser devil ray (EN), and the Ganges shark (*Glyphis gangeticus*, CR; Fig. 1B; Fig. S6D).' – Only four orders are listed but the text mentions five. Also the long fin mako is referred to as the oceanic longfin mako in the text above, this needs to be consistent.

(209) Figure 1 – I think this is a visually attractive figure, but I have a few concerns with it at communicating information to the reader (trait figures in general are often complex and can be difficult to understand).

First, I find the colours very difficult to understand and visually separate. It's very difficult for me to separate the shades of yellow and blue and it is not a colour-blind friendly palette. It's also very difficult to understand how the colours relate across the panels. Are the red and greens in the IUCN plot, or the yellow and blues in the clades plots, the same as the ones in panels B and C? I think more clarity is needed. It seems to me that the most important use of colour is to link the trait space plots for the orders to panel C, so I would focus here on finding the clearest palette of 12 ~separable colours to make this link. This site has some potential colour-blind friendly palettes: <https://personal.sron.nl/~pault/> and here: https://seaborn.pydata.org/tutorial/color_palettes.html#qualitative-color-palettes. I would then avoid using similar colours, or avoid using colour at all, in the rest of panel A or in panel

B. For instance, the clades could be shown in greyscale or just in black as they are already labelled. I think this would also help differentiate the clades from the orders, which might be confusing to the general reader. Same for the IUCN panel where the boxes are already labelled. And the lollipops in panel B could be labelled with CR and EN instead of using colour, or with different shapes on the lollipop (e.g., triangle vs circles), or with colours in greyscale. All of these options would make it harder to think that the colours link across the panels, other than for the orders.

What do the dots represent in panel A? The figure heading doesn't explain. Are they species (but there doesn't seem to be enough dots)? Are they families? I think the dots could just be coloured for the focal order with all other dots on the plot in grey or black. Else, if there is lots of overplotting perhaps contours or 2D density plots would be more informative?

Also, is there no uncertainty associated with the values in panel B? If not, why not, surely these values vary depending on the phylogenetic trees chosen and/or the imputation of missing trait values? I can see why you've just used the model value from the imputations but this hides the fact that these rankings might be very robust to this variance or quite sensitive, which is important when interpreting the rankings. I.e., if conservation prioritized *Mobula tarapacana* this could be well justified or there could be other ~equally important priorities, without including uncertainty it's difficult to tell which is the case.

Finally, I'm not sure it's clear what the main take-homes from Fig. 1 are. Is it that Myliobatiformes cover the greatest breadth of trait space, and Squatiniformes cover the smallest area of trait space (is this the purpose of panel A)? Is it that Lamniformes are the most unique and specialized? Is it that Hexanchiformes are highly unique but Myliobatiformes are highly specialized (if so what does this mean)? I think a bit more work needs to be done so that this figure communicates the main take-homes to the reader. Hopefully, some of my suggestions can help with this, but the authors could also think about how best to display the main messages they want to communicate.

(261) 'We further found elasmobranch mean FUn to display an opposite pattern, concentrating in high latitudes and showing the lowest values along the continental shelf (Fig. 2B).' – could you suggest a reason why Fun shows the opposite pattern to FRic. I think this is worth attention, why do these metrics disagree, what are they prioritizing?

(268) 'The low FUn found where few species co-occur indicates that the system has low redundancy and the loss of individual species is likely to leave large gaps in functional

space14 270.’ – shouldn’t this be high FUn?

(273) Figure 2 – Panels D and E are very similar. Potentially, you could relegate one to the Supplementary and state the similarity (or correlation between the plots) in the text/figure legend. This would give the readers slightly less to digest. Also, for panel C most values appear to be in the same band so it’s hard to see much variation, is this a property of the index? Is there a better way to visualize these values? How come functional uniqueness is mapped as raw numbers but was log-transformed for Figure 1?

(337) ‘found virtually negligible spatial overlap’ – I would suggest deleting ‘virtually’

(344) Figure 3 – again panels B and C are very similar, I’m not sure it is worth presenting both to the reader in the main text.

(502) ‘seven functional traits: body size, habitat, terrestriality, vertical position, diet, feeding mechanism and thermoregulation’ – I think that some of these traits need defining in short here. E.g. “seven functional traits: body size, habitat (coastal, oceanic, or all), terrestriality (marine, brackish, or freshwater), vertical position (benthic, pelagic or both), diet (high vertebrates, fish, invertebrates, or plankton; binary and fuzzy), feeding mechanism (macropredators or filter feeders) and thermoregulation (ectothermy or mesothermy)”. Especially as terrestriality is a strange term to apply to fish!

(521) ‘DE IUCN status’ – should be DD IUCN status.

(523) ‘error for diet = 9%’ - what error is this? Mean absolute percentage error?

(530) ‘We repeated the imputations ten times, using 100 random trees used the modal value across replicates as out final trait prediction (Dataset S2).’ – should be “We repeated the imputations ten times, using 100 random trees and used the modal value across replicates as our final trait prediction (Dataset S2).”

(544) ‘FUSE scores were computed based on extinction probabilities²⁸ in 100 years, as provided by their IUCN status⁴⁴.’ I think the reader needs more detail here. Also, the original reference for this coarse conversion is Mooers et al., 2008. There have also been lots of advances in modelling potential extinctions into the future based on IUCN status (see Andermann et al., 2021; Monroe et al., 2019), which is acknowledged in Griffin et al., 2020 but not here. Would an improved method be more suitable?

(560) ‘We further used a binomial GLMs to explore’ – combines singular and plural, should either be “used a binomial GLM” or “used binomial GLMs”

(563) ‘the differences between in ED, FUn and FSp’ – I think ‘in’ should be deleted.

(616) 'which correspond four management types and four governance types' – probably should be "which correspond to four management types and four governance types"

(624) 'It allows to assess whether MPAs' – "It allows us to assess whether MPAs"

(682) 'Violle C, et al. Functional rarity: Tthe ecology of outliers. Trends in Ecology & Evolution 32, 356-367 (2017).' – delete extra t.

Supplementary information

Two Figure S17s.

The Supplementary Information needs further proof-reading, as there are a few typos and mistakes. E.g., 'A "brackish" trait value was assigned to those species that can netter estuaries.'

Reporting Summary

I think it would make more sense to fill out the Reporting Summary under Ecological, evolutionary and environmental sciences, rather than Life sciences. In fact, the newer Nature Reporting Summary ([nature.com/documents/nr-reporting-summary-flat.pdf](https://www.nature.com/documents/nr-reporting-summary-flat.pdf)) form would probably be better suited (I know it would be annoying to redo the form!). But it covers more points, for instance 'Data presentation - Clearly defined error bars are present and what they represent (SD, SE, CI) is noted' – see comment above.

References

Stein, R.W., Mull, C.G., Kuhn, T.S., Aschliman, N.C., Davidson, L.N., Joy, J.B., Smith, G.J., Dulvy, N.K. and Mooers, A.O., 2018. Global priorities for conserving the evolutionary history of sharks, rays and chimaeras. *Nature ecology & evolution*, 2(2), pp.288-298.

Derrick, D.H., Cheok, J. and Dulvy, N.K., 2020. Spatially congruent sites of importance for global shark and ray biodiversity. *PLoS One*, 15(7), p.e0235559.

Lucifora, L.O., García, V.B. and Worm, B., 2011. Global diversity hotspots and conservation priorities for sharks. *PLoS one*, 6(5), p.e19356.

Pimienta, C., Leprieur, F., Silvestro, D., Lefcheck, J.S., Albouy, C., Rasher, D.B., Davis, M., Svenning, J.C. and Griffin, J.N., 2020. Functional diversity of marine megafauna in the Anthropocene. *Science Advances*, 6(16), p.eaay7650.

Toussaint, A., Charpin, N., Brosse, S. and Villéger, S., 2016. Global functional diversity of freshwater fish is concentrated in the Neotropics while functional vulnerability is widespread. *Scientific reports*, 6(1), p.22125.

Auber, A., Waldock, C., Maire, A., Goberville, E., Albouy, C., Algar, A.C., McLean, M.,

Brind'Amour, A., Green, A.L., Tupper, M. and Vigliola, L., 2022. A functional vulnerability framework for biodiversity conservation. *Nature Communications*, 13(1), p.4774.

Mooers, A.Ø., Faith, D.P. and Maddison, W.P., 2008. Converting endangered species categories to probabilities of extinction for phylogenetic conservation prioritization. *PloS one*, 3(11), p.e3700.

Andermann, T., Faurby, S., Cooke, R., Silvestro, D. and Antonelli, A., 2021. iucn_sim: a new program to simulate future extinctions based on IUCN threat status. *Ecography*, 44(2), pp.162-176.

Monroe, M.J., Butchart, S.H., Mooers, A.O. and Bokma, F., 2019. The dynamics underlying avian extinction trajectories forecast a wave of extinctions. *Biology Letters*, 15(12), p.20190633.

Griffin, J.N., Leprieur, F., Silvestro, D., Lefcheck, J.S., Albouy, C., Rasher, D.B., Davis, M., Svenning, J.C. and Pimiento, C., 2020. Functionally unique, specialised, and endangered (FUSE) species: towards integrated metrics for the conservation prioritisation toolbox. *bioRxiv*, pp.2020-05.

I hope these comments help improve the manuscript!

Rob Cooke

Reviewer #1 (Remarks to the Author):

C1. Thank you for the opportunity to review this very interesting paper. The paper is very well written, presented, and I enjoyed reading it. The paper seemingly fits the scope of the journal and promotes the application of function diversity in elasmobranch conservation prioritisation. Overall, I think this is a strong manuscript and, with a few minor adjustments, would be fit for publication. Below, I provide some comments and feedback that I hope the authors find useful.

We thank the reviewer for their positive and constructive comments, which have greatly helped us improve our manuscript.

C2. Line 52 - Reference needed to support this statement/sentence.

We have added some references to support this statement (L59).

Line 58 – Not entirely unexplored, especially given the co-authors publication history. I would recommend downplaying this statement a touch.

We have downplayed this statement (L60).

Line 95 – It is not clear how the two halves of this sentence are linked. Further, I am not sure that the reference given at the end is the most relevant to support these statements, given it focusses on spatial distributions.

Here, we are listing the areas that have been highlighted in previous studies as conservation priorities for elasmobranchs. The first sentence is about hotspots of evolutionary distinctiveness (Stein et al., 2018) and the second is about the areas of congruence between hotspots of evolutionary distinctiveness and endemism (Derrick et al., 2020). We have edited this paragraph to clarify (L83-99).

Line 133 - Given the introduction of the UN High Seas Biodiversity Treaty, it would be interesting to have a direct comparison between hotspots withing and outside EEZs.

Thank you, we agree that this would be interesting. However, as our paper already includes extensive analyses designed to address our *a priori* research questions, we have decided not to pursue this suggested analysis.

Line 118 – The mismatch they refer to here should be made clearer.

We have clarified that the mismatch refers to rarity hotspots and MPAs (L117-121).

Line 164 – Could the authors double check the FRic scores here. Visually, shark trait space looks larger than batoids in Figure 1.

We have checked our originally reported results and confirmed they were correct. The visual differences were therefore a result of the fact that the elasmobranch trait-space is three-dimensional, with the figure only showing two dimensions. We have clarified this in the revised manuscript. However, based on the reviewer's comment about our reliance on FishBase below, we have updated our trait dataset, which slightly changed our results. Now, sharks occupy a larger extent of the trait-space than batoids. This is now more evident in Fig. 1.

Line 252 – Very little justification is given for the decision to focus on only the top 2.5% cell values,

especially for fishing pressure. This seems to be somewhat selective and based on an arbitrary threshold. Why did the authors adopt this approach, rather than using them as continuous variables instead?

We followed previous studies in the fields of conservation biology and biogeography that define biodiversity hotspots as the 2.5% top values (e.g., Orme et al. 2005; Mouillot et al. 2011; Albouy et al. 2017; Stein et al. 2018). We have added these references to the Methods section of our revised manuscript to clarify. Nevertheless, we recognise that this threshold value is arbitrary, especially for metrics not related to biodiversity. We therefore performed a congruence analysis between the biodiversity hotspots and the fishing pressure index and MPAs using different threshold values, i.e. 2.5%, 5% and 10% (see Figs. S12 and S18).

General point – There is a heavily reliance on Fish Base derived data to inform the trait database. However, some of the data in Fish Base is questionable and coverage is patchy. The authors do a good job in plugging gaps from the known literature but could have gone further with sourcing primary data to inform their trait database. For example, the recent study in the vertical distributions of elasmobranchs (see Andrzejaczek et al 2022). This is pertinent as the vertical position categories are too broad and more refinement/bins would better link to function. Admittedly, it may not be possible to entirely re-run the analyses. However, the limitations of taking a broad-brush approach should be addressed in the manuscript.

Andrzejaczek, et al. 2022. Diving into the vertical dimension of elasmobranch movement ecology. *Science Advances*, 8(33), p.eabo1754.

We thank the reviewer for this important comment. We have checked the trait data of each species in our dataset and updated it based on recent literature while also citing all references in the data file. This update resulted in minor changes in some of the diversity indices. However, the general patterns remain the same. We maintained the broad trait categorisations, as it allows us to assign trait values to as many species as possible, enabling us to use a global-scale approach in our analyses. This is now stated in the Methods section (L545-546).

Line 501 – What was the justification for including only these variables and not others, such as swim speed, trophic position (instead of diet which, looking at Table S1, doesn't contain much variability), a measure of horizontal distribution, or thermal tolerance (more than ecto vs endo).

We selected the traits that characterize elasmobranch functional diversity based on a previous study where each trait is linked to individual- and ecosystem-level functions (see Pimiento et al., 2020; Table S1), while tailoring the trait-assignments to elasmobranchs. We avoided the use of traits that are not well-studied for most species (e.g., swimming speed, migration, breeding site, etc.) to reduce the amount of missing data and the uncertainty around trait-imputations. We excluded trophic position as it would be highly redundant given our use of diet (prey items), which shows great variation among species (Fig. S1E). We have clarified this in the Methods section (L543-544).

Line 548 – The authors note that all traits were weighted equally within the analysis. However, the implications of this assumption are not discussed within the manuscript. As with other comments here, more discussion on the limitations of their general approach would be welcome in the discussions.

We now clarify that we treat all traits equally (assumed equal importance) as we do not have *a priori* reasons to up- or downweight particular traits. On a more technical note, to ensure that traits are indeed equally weighted, care was needed to avoid up-weighting traits with more categories than

others. Accordingly, we used the modified Gower's index by Pavoine et al. (2010), in which each trait displays the same weight in the final calculation of the distance between species, whatever the number of modalities (or categories) of the trait, namely a trait with 8 modalities will have the same weight (importance) that a trait with 2. This is best practise when quantifying functional diversity based on trait distances between species, and avoids giving more importance to traits with more categories and consequently more chance to characterize differences among species. We have clarified this in the Methods section (L580-590).

Line 554 - There has recently been an update EDGE approach, referred to as EDGE2. Given the lack of congruence, it would be interesting to see if this changes much in the analyses. At the very least, the shortcomings of the old approach and the potential of this new metric should be discussed within the paper. <https://journals.plos.org/plosbiology/article?id=10.1371/journal.pbio.3001991>

We thank the reviewer for this comment. Indeed, the EDGE2 approach is a great update to the original metric, as it incorporates uncertainty in the ED calculation and probabilities of extinction over time in the GE calculation. In light to this comment, we have now re-done all our ED calculations using ED2 proposed in Gumbs et al. (2023). However, we did not need to change our GE calculation, as it was originally done using probabilities of extinction in 100 years following Mooers et al. (2008). The congruence between FUSE and EDGE2 was slightly reduced after this change, as it is now 67% instead of 77%. Please note we are no longer comparing ED with FUn, as we have now computed a more comparable index: PUn (Phylogenetic diversity; the mean phylogenetic distance of species to their closest neighbour among co-occurring species in each cell; for more details; please see our response to **Reviewer #2**). The spatial congruence between FUn and PUn is moderate (26%), in contrast with the congruence between FUn and ED (as previously calculated), which was almost negligible (< 1%) in the previous version of the manuscript.

Lines 558 – More information is needed to demonstrate that the statistical approaches are valid (i.e., normally distributed)

Thanks for bringing this up. We have checked and modified accordingly (see L621-632).

Line 583 – Could the authors provide more information on how they summed these different fishing types, many of which have very different units of effort associated with them (i.e., days fished, number of hooks, set time, etc).

To compute our fishing effort index, we summed the hours spent at sea of vessels using fishing types known to target elasmobranch species. We have included this and more details on this index in the revised manuscript (Methods section, L668-685).

Line 583 – Also, were data obtained at the same spatial resolution at the SR calculations? If not, what steps were taken to align these.

Thanks for this comment, which allowed us to make important clarifications regarding our methods. While the fishing data was obtained at a 1° resolution, the biodiversity metrics were calculated using a 0.5° resolution. To align these datasets, we applied an ordinary kriging technique, whereby a distance of 3° around each location was chosen for fishing effort interpolation. This distance constraint was utilized to minimize missing values in our fishing pressure index and to align the resolutions of the different datasets (biodiversity metrics and fishing effort). We have included these details in the Methods section of the revised manuscript (L668-685).

Lin 585 – This statement should be supported by a reference or more information. Whilst true to an extent, I think it is unfair to solely point the finger of blame to SE Asia. There are plenty of issues with the GFW dataset and the AIS data source. A more balanced and comprehensive discussion of those should be added here (or in the discussion).

You are right. We have modified the sentence accordingly (L441-447).

Line 617 – I am not familiar with this ‘Designation Field’ but is there a risk that authors removed some MPAs that use such species as umbrella or flagship species to conserve biodiversity more generally? What does the overlap look like if these are included? Also, how many and what spatial coverage was removed by taking this step?

Our decision-making process to select MPAs was based on relevance for elasmobranchs. As such, we excluded those MPAs designated to protect areas that were not relevant to sharks and rays including forest reserves, archaeological parks, botanical parks, special use forests, etc. (see Supplementary file). In addition, we only included MPAs with IUCN categories I to III (i.e., categories Ia, Ib, II, III), all of which are fully protected marine areas. We selected these categories as a means to provide a conservative assessment of potential protection of elasmobranch biodiversity.

The inclusion of the other MPAs (e.g., categories IV to VI) would imply considering not-fully-protected areas where different human activities are allowed. Similarly, including MPAs designated to preserve areas such as archaeological parks or bird reserves would assume that these areas effectively protect sharks, skates or rays; something we do not have information about. Adding these additional MPAs would increase the “protected” cells by ~8,000, logically increasing the level of protection (see figure below, in contrast with Fig. 4B).

We therefore believe that our critical evaluation of MPAs was appropriate for our study and represents a conservative account of the current level of protection of elasmobranchs. Importantly, our approach aligns with evidence suggesting that there is no social or ecological benefits for partially protected areas (e.g., Turnbull et al., 2021).

Figure 1C – What is determining the order of the y-axis here? Would better to order each by x-axis mean in A, then carry that forward in B, C. This could also free up some space to detail the species names.

The order in 1C is being determined by the species values. This descending order of values was used to facilitate the visualization of the pattern we aim to show, allowing the identification of the most important clades. We have clarified this in the caption. However, in light of the reviewer’s comment, we realised that we needed to order this by the median per clade so that this becomes clearer. We have done that in the revised manuscript. Thank you!

The overall narrative of the paper is understandably focussed on elasmobranchs. However, if we are truly to shift to a functional based priority system for conservation action, then is it appropriate to even focus within a taxonomic group at all? Similar or same functions can be realised/provided by species in other, even less related branches of the tree of life (e.g. large teleosts or marine mammals). Thus, the argument could and should be made that a more holistic based approach should be adopted, rather than focussing on elasmobranchs. I would like to see some more discussion on this topic added to the manuscript.

Thanks for bringing this up. We have added a sentence discussing this issue in the Results and Discussion section (L295-300). “It should be noted that, while we focus here on elasmobranchs, an alternative would be to eschew phylogeny and instead examine functional diversity across multiple clades that potentially contribute towards shared ecosystem functions (e.g., Pimiento *et al.*¹⁸ and Waechter *et al.*^{32, 33}). The extent to which species from other phylogenetic groups (e.g., large teleosts, marine mammals) alter global patterns of functional diversity, e.g., by providing redundancy, remains to be investigated.”.

References cited:

Albouy, C., et al. (2017) Multifaceted biodiversity hotspots of marine mammals for conservation priorities. *Diversity and Distributions* 23: 615-626.

Mouillot, D., et al. (2011). Protected and threatened components of fish biodiversity in the Mediterranean Sea. *Current Biology* 21:1044-1050.

Orme, D., et al., (2005) Global hotspots of species richness are not congruent with endemism or threat. *Nature*, 436: 1016-1019.

Stein, R.W., et al. (2018) Global priorities for conserving the evolutionary history of sharks, rays and chimaeras. *Nat Ecol Evol* 2: 288–298.

Turnbull, J.W., Johnston, E.L. and Clark, G.F. (2021), Evaluating the social and ecological effectiveness of partially protected marine areas. *Conservation Biology*, 35: 921-932. <https://doi.org/10.1111/cobi.13677>

Reviewer #2 (Remarks to the Author):

Dear editor, thank you for the opportunity to review the paper "Functional diversity of sharks and rays is highly vulnerable and supported by unique species and locations worldwide.". This is an interesting study, where the authors are putting together multiple facets of biodiversity (taxonomic, functional and phylogenetic diversity), not only at species level (calculating diversity indices for all species individually) but also at assemblage level (here the authors adopted grid cells as alpha diversity and the global pool of species as gamma diversity).

Indeed, the scientific community is constantly showing that global biodiversity conservation will benefit from considering multiple facets of biodiversity for improving conservation and global fisheries management. Another important aspect is to consider that biodiversity indices (taxonomic, functional, or phylogenetic diversity) are scale dependent, several papers showing this relationship are published. Here the authors did not consider this scale dependence. This study would benefit by dividing the global oceans in ecoregions and use those ecoregions as the gamma diversity. This is a major issue, undermining the study interpretations. Please see detailed comments below.

We thank the reviewer for their constructive comments. Please find our responses to each comment below.

Lines 29 – 31. Suggestion: Maybe this can be rewritten as follows - "Conservation priorities are more efficiently applied at species and spatial levels (taxonomic diversity) and considering complementary biodiversity facets such as functional and phylogenetic diversity. Here, we address this challenge by using a trait dataset of > 1,000 species to assess the conservation status of the multiple facets of elasmobranchs biodiversity."

Thanks for this suggestion. However, we would like to keep the focus of our paper on elasmobranchs by starting the abstract with a sentence about their importance.

Lines 58 – 60. Provide reference.

We have added two references (L49-60).

Lines 88 – 92. Use reference number 2 at the end of this statement as well.

Done.

Lines 112 – 116. This is a strong statement based on only one reference looking at two diversity metrics. Ancestral state reconstruction or the fossil record can answer this question, as can clustering

(which would inform ancestral states) or phylogenetic independent contrasts of correlations between traits. However, a mismatch between phylogenetic and functional diversity does not necessarily mean this at all.

We have edited this statement accordingly. We would like to further clarify that here, we are reporting a previous find of no correspondence between phylogenetic and functional diversity in sharks, whereby evolutionary distinct species are not necessarily functionally distinct. Such a result is not unique to sharks, with similar results being found in birds and mammals (Cooke et al., 2020). We are referring to these two metrics because they are of most relevance to our study. The comparison of these two metrics stems from the expectation of phylogenetic diversity being a proxy for functional diversity under the assumption that phylogenetically close species tend to be more ecologically similar because traits are conserved along evolution (Losos, 2008; Münkemüller, Boucher, Thuiller, & Lavergne, 2015; Wiens et al., 2010). However, in the case of adaptive radiations, species may quickly diverge to avoid competition, and thus may be functionally distinct despite their phylogenetic closeness (Schluter, 2000, 1996; Cachera and Le Loc'h 2017). Within Lamniformes for example, closely related species display asymmetrical trait values, including filter-feeder planktivorous and apex predators consuming high-vertebrates, and also ectothermic and mesothermic physiologies. As such, within this clade, species do not tend to be ecologically similar.

Line 146. "Total inertia" - Is it possible to provide any clarification to the reader about what inertia represents in this context?

Total inertia corresponds to the total variance of the Gower's distance matrix, which we have now noted in the text (L148).

Line 149. "Vertical position in the water columns" - It needs to be clarified this is independent of environment, otherwise you would expect a deep-sea shark to be the species with the lowest value on this axis.

We have clarified this in the revised version (L149-164). Vertical position is either benthic or pelagic whereas habitat is either coastal or oceanic (open ocean).

Lines 179 -183. This is a confusing conclusion, as they must have rediversified after the loss of marine reptiles if actually affected by the extinction, and because above the authors claimed there was no conservation of ecology over time in sharks, yet here are positing ecology was conserved for the last 66 million years. Also, the loss of other species would not lead to the group being the most disparate in terms of space occupation - instead the space would just be filled with more species but the convex hull unchanged (or the occupied space would have been even larger in the past). It might affect FUn only, but not Fsp, as the center would be unchanged.

Fossil evidence suggests that lamniforms did not re-diversify after the loss of marine reptiles (Guinot 2016; Condamine et al., 2019). In fact, Lamniformes are now a species-poor clade (with only 15 living species) despite being highly speciose in the Cretaceous (>3,000 species; Condamine et al., 2019). Despite such a sharp decline in species diversity, this clade has maintained high morphological and ecological disparity (Naylor et al. 1997; Underwood 2006; Guinot 2016; Bazzi et al., 2001), as reflected in our results. We agree that species loss within a clade would not lead to the group being the most disparate in the elasmobranch trait space. We have rephrased these sentences to reflect that we are aiming at highlighting the fact that lamniforms are the most unique and specialised order *despite* encompassing only 2% of the total elasmobranch diversity, and that this depleted diversity and high

ecological disparity may be due to the fact that living lamniforms are the representatives of a once much more diverse clade (L178-190).

Line 190. "Occupy almost the full extent (99%) of the functional space." - The functional space shown at the Figure 1 A represents the functional space occupied by all species. That is = 100% of the species included in this study (100% = 1015 species). However, the functional space shown for the IUCN panel (bottom left) does not seem to display the same functional space structure as the other panels (constructed using all species as the other ones). Here red is the functional space occupied by threatened species (631 species) and green is the non-threatened (283 species). We then have 631 + 283 = 914 species, this means that this new functional space shown in this panel does not represent the 1015 species. Therefore, when the authors say that threatened species "occupy almost the full extent (99%) of the functional space" this new percentage given does not refer to the functional space for all species. In fact this new percentage refers to 914 species (1015 - 914 = 101 species less in comparison with the remaining panels). This may lead the reader to not accurate interpretation. Is there a reason why the authors are not using the same functional space structure in this IUCN panel as in the other ones?

We thank the reviewer for this comment, which made us realise there was a mistake in our code, whereby we accidentally excluded some species from this analysis. We have fixed this problem and have recalculated all IUCN Red List status-based indices based on the total number of species. Although the values have slightly changed, the take-home message remains the same: threatened species occupy almost the full extent of the trait space: 98%. (L192-198).

Lines 233 – 239. It is not clear how the conclusion follows from this? The most endangered are coastal species, but is that because there is insufficient data in the EDGE metric for rarely studied species like deep sea sharks, Hexanchiforms, or megamouths? How much of this is decoupling and how much of this is that the EDGE metric just isn't very good for sharks, or based on a bad phylogeny or taxonomy? In other words, is FUSE just a better metric for both ecological and evolutionary distinctiveness overall than the confounded and partially overlapped EDGE (which combines two unrelated metrics)? If so, then a mismatch cannot be taken to mean decoupling of anything.

Our conclusion stems from the following results:

1. We found no correlation between FUn vs. ED and FSp vs. ED (Spearman's $\rho = 0.2$ and 0.1 respectively, $p < 0.05$), as also found in other studies (e.g., Cooke et al., 2020). Please note that when comparing the evolutionary and functional diversity, we do not include GE (which is provided by IUCN status and denotes endangerment in FUSE and EDGE), as that would artificially inflate the correlation between metrics. That is because both the FUSE and EDGE scores are co-determined by GE, whereby highly scored species are necessarily also highly threatened. As such, we do not compare FUSE vs EDGE values.
2. When contrasting the top 20 FUn vs. top ED species and top FSp vs. top ED species, we found no common species.
3. When comparing top 20 FUSE vs. EDGE species, we found only two shared species.
4. When assessing the distribution of top FUSE and EDGE species in trait space, we found segregation, with top EDGE species occupying low values across PCoA axes.

We have heavily edited this paragraph in light of this comment and those made by **Reviewer 3** (L215-231).

Line 242. Maybe start this paragraph reminding the reader that there is two ways of measuring diversity - at species level (as you shown previously, in this case each species receive an index) and at assemblage level (as you are now proposing here in this section, in this case each assemblage receive an index).

We agree. We have added a few sentences accordingly (247-265).

Line 247. "FRic, FUn, FSp, and FUSE" Here the authors are not considering something important. Those metrics are scale dependent, here you are not comparing species from a defined ecological scale, such as Large Marine Ecosystems, Marine Ecoregions or other ecological classification. Instead, you are using the global scale as your gamma diversity and the grid cells as your alpha diversity. You are comparing a species from North Atlantic with a species from South Australia. This study would benefit by dividing the worlds oceans into specific ecological regions (biogeographical/environmental criteria) and then use each of those as your gamma diversity. The suggestion here is:

1 - Divide the worlds ocean into ecoregions.

2 - Calculate the indices at species level and assemblage level for each of those ecoregions separately.

We thank the reviewer for bringing up this very important and valid issue. We see two situations that justify the use of regional species pools as references (gamma diversity) to calculate alpha (and beta) diversity metrics:

1. In the case where the aim of the study is to tease apart the relative influence of biotic interaction, dispersal limitation and environmental filtering in shaping local assemblage structure and composition (e.g., Suarez-Castro et al. 2022 and Mugnai et al. 2022), which is not the aim of our study.
2. In the case where the study aims at identifying biodiversity hotspots within regions (e.g., by dividing each SR, FRic and PD values at a grid-cell-scale by the species richness, functional volume and sum of branch lengths of the regional species pool, respectively), in order to point policy makers and environmental managers to specific localities within the region of interest for local conservation actions (e.g. Mouton et al. 2022). In that context, two cells with different absolute FRic values locally (e.g., one cell in the Central Indo-Pacific region and another in the Mediterranean Sea) may be considered as hotspots when standardizing by the (gamma) diversity of the region.

Identifying biodiversity hotspots for each region independently is an interesting issue, but it goes beyond the scope of our study, which aims at highlighting spatial patterns of elasmobranch biodiversity at the global scale, specifically by (i) identifying regions of the world where elasmobranch biodiversity concentrates and (ii) determining whether these biodiversity hotspots are well covered by MPAs or overlap with human impacts (i.e., fishing) . Therefore, our work intends to help setting global conservation actions towards those regions where threatened elasmobranch biodiversity is found to be greatest.

Overall, we follow a classical approach in the field of conservation biogeography to identify global biodiversity hotspots (e.g., Orme et al. 2005, Albouy et al. 2017 and more recently Tietje et al. 2023) and we prefer to not follow a regional-based approach, which could be done in a future study with different objectives. It is also worth noting that regional-based studies are generally conducted using a finer spatial resolution (e.g., Mouton et al. 2022) and based on the compilation of fieldwork data,

which therefore allows accounting for species abundances in the calculation of the different biodiversity metrics. We now discuss this point in the revised manuscript (L418-425: “While our global analysis represents an important step towards prioritizing conservation actions in the regions where threatened elasmobranch biodiversity is the greatest, it does not allow identifying biodiversity hotspots for each marine biogeographic realm³⁹. Future studies could complement our findings by adopting a regional-based approach, where each region can be considered as independent pool of species. Coupled to finer distribution data at the regional scale (see Mouton et al.⁴⁰) and to long-term abundance trends when available⁴¹, this could ultimately help policy and decision makers, and environmental managers, to set conservation actions towards threatened biodiversity at a regional scale (e.g., Mouillot et al.⁸).”

Importantly, our approach does not completely ignore scale-dependency given that our grid-cell analyses consider only co-occurring species (at the exception of FUn in the original manuscript, see below). Accordingly, in our biogeographic patterns section, we do not compare species from the North Atlantic with species from South Australia. Instead, we re-calculate FRic, FUn and FSp per grid cell (Fig. 2A-C). However, for FUn, we indeed originally calculated this index by considering the nearest neighbour within the global pool of species, and consequently accounted for species from different regions that do not necessarily co-occur (e.g., Violle et al. 2017). In light of the reviewer’s comment, in the revised manuscript, **we have addressed this issue by modifying our calculation of FUn per grid so that instead of it being based on the nearest neighbour within the global pool of species, we are now using the nearest neighbour within co-occurring species.** In that context, FUn reflects the degree to which assemblages are composed, on average, of functionally unique species at the local scale. In other words, this new approach considers the extent to which species co-occurring locally can play a similar functional role if another species were to disappear within this assemblage. We consider that it is also interesting to know, from a conservation perspective, whether assemblages are composed of functionally unique species at the global scale (i.e., composed of species that do not have close relatives in the world’s ocean). When considering this alternative approach, we found that FUn calculated based on co-occurring species or calculated considering all the species the world’s oceans, follow a similar spatial pattern (Figs. 2B and S10E), with FUn increasing towards the poles. This is an important result because as discussed by Violle et al. (2017), assemblages may be composed of functionally unique species locally but not regionally.

Line 268. “The low..”. Is it low or high?

It is the high. Many thanks!

Line 309. “Spiritu Santo” – Espírito Santo.

Done.

Lines 475 – 478. I am not sure what this means, or why it is important. The main axes were habitat, water depth, and diet, but it seems like they use the metrics drawn from their space to make a lot of other inferences about how ecologically unique a species is. There is no comparison of species in similar space or habitat along other traits, which would be required to make this claim, and there is no examination of close relatives.

We consider this to be a key finding of our study as it relates with the highest FUSE species at the global scale. Indeed, we rely on the position of species in trait space to determine how ecologically unique and specialised they are, following widely used methods in functional ecology and regardless

of phylogenetic similarities. Critically, we do take into account species' proximity in the functional space by quantifying functional uniqueness within the FUSE index, which measures species' distance to their nearest neighbours. It is also worth noting that the 3D trait space is based on all of the 8 traits that were used in the calculation of the Gower distance matrix, and therefore, it is not based only on the three traits that most strongly correlate with the axes. Accordingly, metrics drawn from species' positions in the space can be interpreted within the context of the suite of underlying traits.

Lines 487 – 489. Earlier (lines 179 - 183) the authors said that Lamniform diversity is the result of being nearly wiped out by the end-Cretaceous...and that there is no conservation of ecology in sharks. The same statement could be made about tetrapods or teleosts and it would be equally facile. Sharks were also freshwater before the Mesozoic...

Please see our response above regarding the lack of ecological conservatism along the shark tree of life. To summarise, we did not imply that Lamniform diversity is the result of being nearly wiped out by the end-Cretaceous. Instead, we reported previous findings of this clade having a high ecological disparity despite having faced extinctions in the geological past that resulted in their low-species number of today. The lamniform case does not negate the statement of elasmobranchs being a crucial component of marine ecosystems over time, surviving multiple mass extinctions and environmental changes and it does not imply that the same can be said for other clades.

Line 691. "global diversity" – Global diversity.

Fixed.

References cited:

- Albouy, C., et al. (2017) Multifaceted biodiversity hotspots of marine mammals for conservation priorities. *Diversity and Distributions* 23: 615-626.
- Bazzi, M., Campione, N.E., Kear, B.P., Pimiento, C. and Ahlberg, P.E., 2021. Feeding ecology has shaped the evolution of modern sharks. *Current Biology*, 31(23), pp.5138-5148.
- Cachera, M. and Le Loc'h, F., 2017. Assessing the relationships between phylogenetic and functional singularities in sharks (Chondrichthyes). *Ecology and Evolution*, 7(16), pp.6292-6303.
- Condamine FL, Romieu J, Guinot G. Climate cooling and clade competition likely drove the decline of lamniform sharks. *Proceedings of the National Academy of Sciences* 116, 796 20584-20590 (2019).
- Cooke, RSC, Bates, AE, Eigenbrod, F. Global trade-offs of functional redundancy and functional dispersion for birds and mammals. *Global Ecol Biogeogr.* 2019; 28: 484–495.
- Guinot G, Cavin L. 'Fish'(Actinopterygii and Elasmobranchii) diversification patterns through deep time. *Biological Reviews* 91, 950-981 (2016).
- Losos, J. B. (2008). Phylogenetic niche conservatism, phylogenetic signal and the relationship between phylogenetic relatedness and ecological similarity among species. *Ecology letters*, 11(10), 995-1003.
- Mugnai, M., et al. (2022) Environment and space drive the community assembly of Atlantic European grasslands: Insights from multiple facets. *Journal of Biogeography* 49 : 699-711.
- Münkemüller, T., Boucher, F. C., Thuiller, W., & Lavergne, S. (2015). Phylogenetic niche conservatism—common pitfalls and ways forward. *Functional ecology*, 29(5), 627-639.
- Mouton, T.L., et al. (2022) Spatial mismatch in diversity facets reveals contrasting protection for New Zealand's cetacean biodiversity. *Biological Conservation* 267: 109484.

Naylor GJ, Caira JN, Jensen K, Rosana KA, Straube N, Lakner C. Elasmobranch phylogeny: a mitochondrial estimate based on 595 species. *The biology of sharks and their relatives*, 31-56 (2012).

Orme, D., et al., (2005) Global hotspots of species richness are not congruent with endemism or threat. *Nature* 436: 1016-1019.

Schluter, D. (2000). Ecological character displacement in adaptive radiation. *The American Naturalist*, 156(S4), S4-S16

Suarez-Castro, A-F., et al. (2022) Using multi-scale spatially explicit frameworks to understand the relationship between functional diversity and species richness. *Ecography* 6 : e05844

Tietje, M., et al. (2023) Global hotspots of plant phylogenetic diversity. *New Phytologist* : doi: 10.1111/nph.19151.

Underwood, C.J., 2006. Diversification of the Neoselachii (Chondrichthyes) during the Jurassic and Cretaceous. *Paleobiology*, 32(2), pp.215-235.

Violle, C., et al. (2017). Functional rarity: The ecology of outliers. *Trends in Ecology & Evolution* 32(5): 356–367.

Wiens, J.J., Ackerly, D.D., Allen, A.P., Anacker, B.L., Buckley, L.B., Cornell, H.V., Damschen, E.I., Jonathan Davies, T., Grytnes, J.-A., Harrison, S.P., Hawkins, B.A., Holt, R.D., McCain, C.M. and Stephens, P.R. (2010), Niche conservatism as an emerging principle in ecology and conservation biology. *Ecology Letters*, 13: 1310-1324.

Reviewer #3 (Remarks to the Author):

The authors analyse the functional diversity of sharks and rays (~1,100 species) across the globe using multiple functional diversity metrics (functional richness, functional uniqueness, functional specialization), as well as comparing functional diversity to other biodiversity facets (species richness, phylogenetic diversity, evolutionary distinctiveness). They find that threatened species disproportionately contribute to functional diversity. They also identify multiple unique elasmobranch functional diversity spatial hotspots (not represented by other biodiversity facets), including around oceanic islands and high seas. Yet much of this elasmobranch diversity is unprotected through the global MPA network. Overall, I think this is an interesting study that covers a lot of ground in terms of biodiversity metrics and spatial prioritization. Even though there are a growing number of papers on this topic (Stein et al., 2018; Derrick et al., 2020; Lucifora et al., 2011; Pimiento et al., 2020), I think it is a useful addition to the literature. In particular, I think the authors have done a particularly good job of presenting multiple metrics of biodiversity and multiple functional diversity metrics for all elasmobranch species (previous papers generally don't cover all elasmobranch species, or, where they do, they generally focus on taxonomic and phylogenetic diversity, or single metrics of functional diversity). I also think that the manuscript is well articulated and the analyses generally well executed. Still, I have quite a few comments, which I have listed below (line number in brackets). To be clear, these are suggestions and I leave these up to the discretion of the authors who will know their study much more closely than I do.

We thank the reviewer for their comments and useful suggestions.

(30) 'assess elasmobranch functional diversity' – personally, I think it would be informative for the reader to know which metrics of functional diversity have been used, as functional diversity means different things to different people. Perhaps: "assess elasmobranch functional diversity (functional richness, functional uniqueness, and functional specialization)".

Thanks for this suggestion. Given the word-limit of the abstract, we would like to keep this as it is because this addition would require the description of a number of metrics, significantly increasing the length of the abstract.

(30) 'compare it against other previously studied biodiversity facets' – again, it is unclear what other biodiversity facets were investigated from the abstract alone. I understand that space is limited in an abstract, but I think this information would be really useful for the reader to set the analysis in context.

We have added 'taxonomic and phylogenetic' to specify the other facets (L32). However, we would require much more text to fully explain this than is available in the short abstract, since we compared specific aspects of functional diversity (e.g., FUn) against specific corresponding aspects of diversity in other dimensions (e.g., phylogenetic: PUn). Accordingly, while we agree that it would be desirable to specify the exact metrics of functional diversity we use, we have opted to leave it general to avoid too much detail and to retain flow.

(44) 'elevated functional vulnerability of the world's sharks and rays' – vulnerability is not mentioned in the main text. I also assume that the authors mean vulnerability in the general sense, rather than referring to more explicit implementations to functional vulnerability (e.g., Toussaint et al., 2016; Auber et al., 2022). I think it needs to be much clearer in the main text what is being referred to by the term functional vulnerability, or vulnerability should be dropped or reworded in the abstract.

We agree this was confusing. We were trying to refer to 'vulnerability' in the true sense of the word (Oxford definition: "the quality or state of being exposed to the possibility of being attacked or harmed"). Given our results showing high overlap between functionally rich areas and fishing pressure, among others, we think this is a well-justified word that captures the essence of our findings. We have reordered the words in the abstract to avoid conflating this with the more specific and technical 'functional vulnerability' used in the functional diversity literature (L44).

(58) 'Nevertheless, for elasmobranchs, global diversity and prioritisation studies have focused mainly on the evolutionary component' – for me this needs referencing, what studies are you referring to? We have added two references.

(124) 'Here, we assemble a trait data set for over 1,000 elasmobranch species to assess their functional diversity.' – I think it would be good to include a percentage here for how many of the total species are included. E.g. "Here, we assemble a trait data set for over 1,000 elasmobranch species (X% of all elasmobranch species) to assess their functional diversity."

Done (L124)

(126) 'We then quantify the contributions of individual species to functional diversity and apply the FUSE (Functionally Unique, Specialised, and Endangered) conservation prioritization metric^{14 128}, to identify highly threatened species whose extinction would result in the most significant functional losses.' – I would be included to change this to "... whose global extinction ...", just to be clear. As when considering local or regional extinctions the importance of species may be different (different context of extinction).

Done.

(142) 'assigned seven functional traits' – I think it would be useful to specify which traits were used here.

In this Results and Discussion section, we would like to provide a broad picture of our methodological approach while focusing on our findings. Therefore, we have expanded on the functional traits in the methods section and added "see methods" here (L142-144).

(159) 'FRic = % volume of the trait space occupied'. What does this volume refer to? How was this volume calculated? Volume is not mentioned in the Methods and it is unclear exactly how this metric was calculated. As there are many ways to calculate the volume of a trait space available (e.g., hypervolumes, convex hull, kernels) I think more detail is needed here/in the Methods.

We used convex hull volumes. We have clarified this in the Methods section (L579-610).

(169) 'with members ranging a wide variety of sizes, habitats and diets' – probably should be "with members ranging across a wide variety of sizes, habitats and diets"

Done.

(170) 'As such, stingrays have the most extreme trait-combinations (FSp) within their own functional space (Fig. 1A; Table S3).' – what do you mean by their 'own' functional space? Aren't all species compared across the same trait space, whereas this wording implies that each group had a separate trait space built? Perhaps, "As such, stingrays have the most extreme trait-combinations (FSp) across the elasmobranch functional space (Fig. 1A - Myliobatiformes; Table S3)." I also added the order name again just to make it clear what part of Fig 1A is being referred to here.

Thanks for pointing out this inconsistency. We have heavily edited this section, using the above-mentioned suggestion to clarify. Note that given the changes in our data (see our responses to **Reviewer #1**), we now found Lamniformes to be the most specialised and unique order (166-190).

(199) 'Nevertheless, the top five FUSE species span five orders (Myliobatiformes, Rhinopristiformes, Lamniformes and Carcharhiniformes) and include the Chilean devil ray (*Mobula tarapacana*, EN), the broad-nose wedge-fish (*Rhynchobatus springeri*, CR), the long fin mako (EN), the lesser devil ray (EN), and the Ganges shark (*Glyphis gangeticus*, CR; Fig. 1B; Fig. S6D)." – Only four orders are listed but the text mentions five. Also the long fin mako is referred to as the oceanic longfin mako in the text above, this needs to be consistent.

We corrected the number of orders to four. The longfin mako is only referred as such, with the word oceanic referring to its habitat and not its common name. We have added "open ocean" for clarification.

(209) Figure 1 – I think this is a visually attractive figure, but I have a few concerns with it at communicating information to the reader (trait figures in general are often complex and can be difficult to understand).

First, I find the colours very difficult to understand and visually separate. It's very difficult for me to separate the shades of yellow and blue and it is not a colour-blind friendly palette. It's also very difficult to understand how the colours relate across the panels. Are the red and greens in the IUCN plot, or the yellow and blues in the clades plots, the same as the ones in panels B and C? I think more clarity is needed. It seems to me that the most important use of colour is to link the trait space plots for the orders to panel C, so I would focus here on finding the clearest palette of 12 ~separable

colours to make this link. This site has some potential colour-blind friendly palettes: <https://personal.sron.nl/~pault/> and

here: https://seaborn.pydata.org/tutorial/color_palettes.html#qualitative-color-palettes. I would then avoid using similar colours, or avoid using colour at all, in the rest of panel A or in panel B. For instance, the clades could be shown in greyscale or just in black as they are already labelled. I think this would also help differentiate the clades from the orders, which might be confusing to the general reader. Same for the IUCN panel where the boxes are already labelled.

And the lollipops in panel B could be labelled with CR and EN instead of using colour, or with different shapes on the lollipop (e.g., triangle vs circles), or with colours in greyscale. All of these options would make it harder to think that the colours link across the panels, other than for the orders.

What do the dots represent in panel A? The figure heading doesn't explain. Are they species (but there doesn't seem to be enough dots)? Are they families? I think the dots could just be coloured for the focal order with all other dots on the plot in grey or black. Else, if there is lots of overplotting perhaps contours or 2D density plots would be more informative?

Also, is there no uncertainty associated with the values in panel B? If not, why not, surely these values vary depending on the phylogenetic trees chosen and/or the imputation of missing trait values? I can see why you've just used the model value from the imputations but this hides the fact that these rankings might be very robust to this variance or quite sensitive, which is important when interpreting the rankings. I.e., if conservation prioritized *Mobula tarapacana* this could be well justified or there could be other ~equally important priorities, without including uncertainty it's difficult to tell which is the case.

Finally, I'm not sure it's clear what the main take-homes from Fig. 1 are. Is it that Myliobatiformes cover the greatest breadth of trait space, and Squatiniformes cover the smallest area of trait space (is this the purpose of panel A)? Is it that Lamniformes are the most unique and specialized? Is it that Hexanchiformes are highly unique but Myliobatiformes are highly specialized (if so what does this mean)? I think a bit more work needs to be done so that this figure communicates the main take-homes to the reader. Hopefully, some of my suggestions can help with this, but the authors could also think about how best to display the main messages they want to communicate.

We thank the reviewer for this through evaluation of our Fig. 1. Please find below our responses and actions (in bold):

- Purpose: The purpose of this figure is to show functional diversity across clades and IUCN Red List status, which is the focus of the first section of our results. This figure therefore shows that 1) myliobatiforms span the largest portion of trait space; 2) lamniforms is the most functionally unique and specialised order (with species sitting at extreme ends of the space and far apart from each other); 3) that endangered species occupy the full extent of trait space; 4) the ranking of the top 12 FUSE species; and the 5) the ranking of FUn, FSp and FUSE per order.
- Colours: Colours are used to denote the different clades and level of endangerment, whereby each clade and level of endangerment is represented by a unique colour. These colours are consistent throughout the figure; however, we use transparency inside the spaces to allow seeing overlapping elements. Regarding the IUCN plots, we used standard colours (we actually used the IUCN colour palette in R), whereby endangered and not-endangered categories are represented by green and red in the functional space plot, and endangered (EN) and critically endangered (CR) by orange and red in the FUSE plot.

- Functional spaces: Each dot inside the space represent species, with great overlap. Based on the reviewer's comment, **we have clarified this on the figure label**. Furthermore, **we have changed the colours of the spaces**, by using grey to denote all species while highlighting only the focal order using a colour-blind-friendly palette of 12 qualitative colours, as suggested. Please note that given the difficulty to find 12 totally different colours, even after our changes there are still some colours that are quite similar. However, we consider that our approach of using colours, labels and animal shapes greatly facilitates interpretability. **We have further divided Fig 1A in two (A and B) to enhance clarity**, with A being the spaces of superorders and IUCN Red List status, and B of orders.
- FUSE: **We are now including a legend inside the plot to clarify** what the colours mean. Finally, **we have addressed the issue on uncertainty around trait values by including the following additions to the revised manuscript**:
 - o We compared inferred trait values across the 10 imputations: This allowed us to demonstrate the low variability found across iterations, which is depicted in a correlogram (new Fig. S19).
 - o We computed the FUSE metric using each imputed dataset and calculated the mean and standard deviation across imputations. As anticipated given high consistency and correlation across imputed data (Fig. S19), we found very small variation around FUSE values (mean standard variation = 0.00241) suggesting that our ranking of top FUSE species is robust. We have included a plot in the revised manuscript showing the standard variation of FUSE values for top 20 species (Fig. S6F).

(261) 'We further found elasmobranch mean FUn to display an opposite pattern, concentrating in high latitudes and showing the lowest values along the continental shelf (Fig. 2B).' – could you suggest a reason why Fun shows the opposite pattern to FRic. I think this is worth attention, why do these metrics disagree, what are they prioritizing?

Thanks for this comment. One reason why FUn shows the opposite pattern to FRic is because these metrics have diametrically opposed associations with species richness. As such, FRic (i.e., volume of trait space) is positively associated with species richness whereas FUn (i.e., species' distance to nearest neighbours) displays the inverse pattern. In our case, these relationships are triangular, whereby both low and high FRic values can be found in species-rich assemblages, a pattern also revealed for marine mammals (Albouy et al. 2017). This implies some species-poor assemblages are composed of species that have dissimilar traits (with less scope for functional redundancy), but other assemblages contain few species with relatively similar traits (low FUn) and therefore are functionally redundant. We have clarified this in the revised manuscript (L267-300), including a new plot that shows the relation between FUn (and FSp) vs. species richness (Fig. S9).

Also, please note that we have stated in our manuscript that isolated areas with few species that are functionally unique and specialised should be considered as priority areas for conservation.

(268) 'The low FUn found where few species co-occur indicates that the system has low redundancy and the loss of individual species is likely to leave large gaps in functional space¹⁴ 270.' – shouldn't this be high FUn?

Yes. We have corrected this. Thanks!

(273) Figure 2 – Panels D and E are very similar. Potentially, you could relegate one to the Supplementary and state the similarity (or correlation between the plots) in the text/figure legend.

This would give the readers slightly less to digest. Also, for panel C most values appear to be in the same band so it's hard to see much variation, is this a property of the index? Is there a better way to visualize these values? How come functional uniqueness is mapped as raw numbers but was log-transformed for Figure 1?

We agree. We have modified this figure, relegating some plots to the supplement Figs S10-S12). We have also modified the colours of the FSp map to improve visualisation. Indeed, most of the FSp values are close to the global median, which results in a rather homogenous pattern. We have reworded the way we present this result to make this clear (L285-288).

FUn is log-transformed in Fig. 1 to facilitate visualization given the skewness of the raw values (see figure below). This, however, does not represent an issue when mapping the distribution across grid cells given that we plot mean values per cell and show them using a colour gradient. We have added a brief line on this in the Figure 1's caption.

(337) 'found virtually negligible spatial overlap' – I would suggest deleting 'virtually' Done.

(344) Figure 3 – again panels B and C are very similar, I'm not sure it is worth presenting both to the reader in the main text.

We have modified this figure accordingly.

(502) 'seven functional traits: body size, habitat, terrestriality, vertical position, diet, feeding mechanism and thermoregulation' – I think that some of these traits need defining in short here. E.g. "seven functional traits: body size, habitat (coastal, oceanic, or all), terrestriality (marine, brackish, or freshwater), vertical position (benthic, pelagic or both), diet (high vertebrates, fish, invertebrates, or plankton; binary and fuzzy), feeding mechanism (macro predators or filter feeders) and thermoregulation (ectothermy or mesothermy)". Especially as terrestriality is a strange term to apply to fish!

We agree and have added these details (L531-546).

(521) 'DE IUCN status' – should be DD IUCN status.

Done.

(523) 'error for diet = 9%' - what error is this? Mean absolute percentage error?

The error was calculated as follows: number of incorrect / total number of imputed * 100. We have clarified this in the Methods section (L561-577).

(530) 'We repeated the imputations ten times, using 100 random trees used the modal value across replicates as our final trait prediction (Dataset S2).' – should be "We repeated the imputations ten times, using 100 random trees and used the modal value across replicates as our final trait prediction (Dataset S2)."

Done.

(544) 'FUSE scores were computed based on extinction probabilities²⁸ in 100 years, as provided by their IUCN status⁴⁴.' I think the reader needs more detail here. Also, the original reference for this coarse conversion is Mooers et al., 2008. There have also been lots of advances in modelling potential extinctions into the future based on IUCN status (see Andermann et al., 2021; Monroe et al., 2019), which is acknowledged in Griffin et al., 2020 but not here. Would an improved method be more suitable?

We have used the probabilities of extinction proposed by Mooers et al., (2008) given that these are also used in EDGE2 framework (Gumbs et al., 2023), to ensure a consistent comparison. We have clarified this in the revised manuscript.

(560) 'We further used a binomial GLMs to explore' – combines singular and plural, should either be "used a binomial GLM" or "used binomial GLMs".

Fixed.

(563) 'the differences between in ED, FUn and FSp' – I think 'in' should be deleted.

Thank you. We have edited this paragraph and this sentence is no longer included as it was originally stated.

(616) 'which correspond four management types and four governance types' – probably should be "which correspond to four management types and four governance types"

Fixed.

(624) 'It allows to assess whether MPAs' – "It allows us to assess whether MPAs"

Done.

(682) 'Violle C, et al. Functional rarity: The ecology of outliers. Trends in Ecology & Evolution 32, 356-367 (2017).' – delete extra t

Fixed.

Supplementary information

Two Figure S17s.

We have corrected this, thank you.

The Supplementary Information needs further proof-reading, as there are a few typos and mistakes. E.g., 'A "brackish" trait value was assigned to those species that can netter estuaries.'

We have proofread the document, fixing all errors and typos.

Reporting Summary

I think it would make more sense to fill out the Reporting Summary under Ecological, evolutionary and environmental sciences, rather than Life sciences. In fact, the newer Nature Reporting Summary ([nature.com/documents/nr-reporting-summary-flat.pdf](https://www.nature.com/documents/nr-reporting-summary-flat.pdf)) form would probably be better suited (I know it would be annoying to redo the form!). But it covers more points, for instance 'Data presentation - Clearly defined error bars are present and what they represent (SD, SE, CI) is noted' – see comment above.

We thank the reviewer for his comment. We have submitted a new form (however, please see our answer to the comment regarding uncertainty around FUSE values, which did not result in the addition of error bars).

References

- Stein, R.W., Mull, C.G., Kuhn, T.S., Aschliman, N.C., Davidson, L.N., Joy, J.B., Smith, G.J., Dulvy, N.K. and Mooers, A.O., 2018. Global priorities for conserving the evolutionary history of sharks, rays and chimaeras. *Nature ecology & evolution*, 2(2), pp.288-298.
- Derrick, D.H., Cheok, J. and Dulvy, N.K., 2020. Spatially congruent sites of importance for global shark and ray biodiversity. *PLoS One*, 15(7), p.e0235559.
- Lucifora, L.O., García, V.B. and Worm, B., 2011. Global diversity hotspots and conservation priorities for sharks. *PLoS one*, 6(5), p.e19356.
- Pimiento, C., Leprieur, F., Silvestro, D., Lefcheck, J.S., Albouy, C., Rasher, D.B., Davis, M., Svenning, J.C. and Griffin, J.N., 2020. Functional diversity of marine megafauna in the Anthropocene. *Science Advances*, 6(16), p.eaay7650.
- Toussaint, A., Charpin, N., Brosse, S. and Villéger, S., 2016. Global functional diversity of freshwater fish is concentrated in the Neotropics while functional vulnerability is widespread. *Scientific reports*, 6(1), p.22125.
- Auber, A., Waldock, C., Maire, A., Goberville, E., Albouy, C., Algar, A.C., McLean, M., Brind'Amour, A., Green, A.L., Tupper, M. and Vigliola, L., 2022. A functional vulnerability framework for biodiversity conservation. *Nature Communications*, 13(1), p.4774.
- Mooers, A.Ø., Faith, D.P. and Maddison, W.P., 2008. Converting endangered species categories to probabilities of extinction for phylogenetic conservation prioritization. *PloS one*, 3(11), p.e3700.
- Andermann, T., Faurby, S., Cooke, R., Silvestro, D. and Antonelli, A., 2021. iucn_sim: a new program to simulate future extinctions based on IUCN threat status. *Ecography*, 44(2), pp.162-176.
- Monroe, M.J., Butchart, S.H., Mooers, A.O. and Bokma, F., 2019. The dynamics underlying avian extinction trajectories forecast a wave of extinctions. *Biology Letters*, 15(12), p.20190633.
- Griffin, J.N., Leprieur, F., Silvestro, D., Lefcheck, J.S., Albouy, C., Rasher, D.B., Davis, M., Svenning, J.C. and Pimiento, C., 2020. Functionally unique, specialised, and endangered (FUSE) species: towards integrated metrics for the conservation prioritisation toolbox. *bioRxiv*, pp.2020-05.

I hope these comments help improve the manuscript!

Rob Cooke

References cited:

Albouy, C., et al. (2017) Multifaceted biodiversity hotspots of marine mammals for conservation priorities. *Diversity and Distributions* 23: 615-626.

REVIEWERS' COMMENTS

Reviewer #1 (Remarks to the Author):

I think the authors have done a sufficient job in addressing my feedback and comments in this updated manuscript. As such, I am now satisfied and recommend the paper be accepted.

Reviewer #2 (Remarks to the Author):

Line 247. "FRic, FUn, FSp, and FUSE" Here the authors are not considering something important. Those metrics are scale dependent, here you are not comparing species from a defined ecological scale, such as Large Marine Ecosystems, Marine Ecoregions or other ecological classification. Instead, you are using the global scale as your gamma diversity and the grid cells as your alpha diversity. You are comparing a species from North Atlantic with a species from South Australia. This study would benefit by dividing the worlds oceans into specific ecological regions (biogeographical/environmental criteria) and then use each of those as your gamma diversity.

The suggestion here is:

- 1 - Divide the worlds ocean into ecoregions.
- 2 - Calculate the indices at species level and assemblage level for each of those ecoregions separately.

We thank the reviewer for bringing up this very important and valid issue. We see two situations that justify the use of regional species pools as references (gamma diversity) to calculate alpha (and beta) diversity metrics:

1. In the case where the aim of the study is to tease apart the relative influence of biotic interaction, dispersal limitation and environmental filtering in shaping local assemblage structure and composition (e.g., Suarez-Castro et al. 2022 and Mugnai et al. 2022), which is not the aim of our study.
2. In the case where the study aims at identifying biodiversity hotspots within regions (e.g., by dividing each SR, FRic and PD values at a grid-cell-scale by the species richness, functional volume and sum of branch lengths of the regional species pool, respectively), in order to point policy makers and environmental managers to specific localities within the region of

interest for local conservation actions (e.g. Mouton et al. 2022). In that context, two cells with different absolute FRic values locally (e.g., one cell in the Central Indo-Pacific region and another in the Mediterranean Sea) may be considered as hotspots when standardizing by the (gamma) diversity of the region.

Identifying biodiversity hotspots for each region independently is an interesting issue, but it goes beyond the scope of our study, which aims at highlighting spatial patterns of elasmobranch biodiversity at the global scale, specifically by (i) identifying regions of the world where elasmobranch biodiversity concentrates and (ii) determining whether these biodiversity hotspots are well covered by MPAs or overlap with human impacts (i.e., fishing). Therefore, our work intends to help setting global conservation actions towards those regions where threatened elasmobranch biodiversity is found to be greatest.

Overall, we follow a classical approach in the field of conservation biogeography to identify global biodiversity hotspots (e.g., Orme et al. 2005, Albouy et al. 2017 and more recently Tietje et al. 2023) and we prefer to not follow a regional-based approach, which could be done in a future study with different objectives. It is also worth noting that regional-based studies are generally conducted using a finer spatial resolution (e.g., Mouton et al. 2022) and based on the compilation of fieldwork data, which therefore allows accounting for species abundances in the calculation of the different biodiversity metrics. We now discuss this point in the revised manuscript (L418-425: “While our global analysis represents an important step towards prioritizing conservation actions in the regions where threatened elasmobranch biodiversity is the greatest, it does not allow identifying biodiversity hotspots for each marine biogeographic realm³⁹. Future studies could complement our findings by adopting a regional-based approach, where each region can be considered as independent pool of species. Coupled to finer distribution data at the regional scale (see Mouton et al.⁴⁰) and to long-term abundance trends when available⁴¹, this could ultimately help policy and decision makers, and environmental managers, to set conservation actions towards threatened biodiversity at a regional scale (e.g., Mouillot et al.⁸).”

Importantly, our approach does not completely ignore scale-dependency given that our grid-cell analyses consider only co-occurring species (at the exception of FUn in the original

manuscript, see below). Accordingly, in our biogeographic patterns section, we do not compare species from the North Atlantic with species from South Australia. Instead, we recalculate FRic, FUn and FSp per grid cell (Fig. 2A-C). However, for FUn, we indeed originally calculated this index by considering the nearest neighbour within the global pool of species, and consequently accounted for species from different regions that do not necessarily co-occur (e.g., Violle et al. 2017). In light of the reviewer's comment, in the revised manuscript, we have addressed this issue by modifying our calculation of FUn per grid so that instead of it being based on the nearest neighbour within the global pool of species, we are now using the nearest neighbour within co-occurring species. In that context, FUn reflects the degree to which assemblages are composed, on average, of functionally unique species at the local scale. In other words, this new approach considers the extent to which species co-occurring locally can play a similar functional role if another species were to disappear within this assemblage. We consider that it is also interesting to know, from a conservation perspective, whether assemblages are composed of functionally unique species at the global scale (i.e., composed of species that do not have close relatives in the world's ocean). When considering this alternative approach, we found that FUn calculated based on co-occurring species or calculated considering all the species the world's oceans, follow a similar spatial pattern (Figs. 2B and S10E), with FUn increasing towards the poles. This is an important result because as discussed by Violle et al. (2017), assemblages may be composed of functionally unique species locally but not regionally.

Reviewer answer: I would like to thank the authors for providing further clarifications between lines L418-425 and for addressing the scale issue pointed in the previous review. The modification of the method applied, from using the nearest neighbour within the global pool of species to using the nearest neighbour within co-occurring species, provides a valid alternative to address the scale dependence issue.

Line 190. "Occupy almost the full extent (99%) of the functional space." - The functional space shown at the Figure 1 A represents the functional space occupied by all species. That is = 100% of the species included in this study (100% = 1015 species). However, the functional space shown for the IUCN panel (bottom left) does not seem to display the same functional space structure as the other panels (constructed using all species as the other

ones). Here red is the functional space occupied by threatened species (631 species) and green is the non-threatened (283 species). We then have $631 + 283 = 914$ species, this means that this new functional space shown in this panel does not represent the 1015 species. Therefore, when the authors say that threatened species "occupy almost the full extent (99%) of the functional space" this new percentage given does not refer to the functional space for all species. In fact this new percentage refers to 914 species ($1015 - 914 = 101$ species less in comparison with the remaining panels). This may lead the reader to not accurate interpretation. Is there a reason why the authors are not using the same functional space structure in this IUCN panel as in the other ones?

We thank the reviewer for this comment, which made us realise there was a mistake in our code, whereby we accidentally excluded some species from this analysis. We have fixed this problem and have recalculated all IUCN Red List status-based indices based on the total number of species. Although the values have slightly changed, the take-home message remains the same: threatened species occupy almost the full extent of the trait space: 98%. (L192-198).

Reviewer answer: Thanks for addressing this point.

Reviewer #3 (Remarks to the Author):

I previously reviewed this manuscript and believe the authors have done a good job at revising the manuscript. I really like the new and improved Fig. 1 and I think the additional trait imputation analyses give much more confidence to the reader about the robustness of the prioritization results. Overall, I believe the updated manuscript is much improved and I only have one remaining comment.

(References) The references probably need double-checking as there are a few weird issues/formatting errors.

For instance, reference 27 "union wc. IUCN Red list categories and criteria: version 3.1. IUCN Species survival commission.). IUCN Gland^ eU. KSwitzerlandCambridge UK (2001)."

Reference 28 "Guinot G, Cavin L. 'Fish'(Actinopterygii and E lasmobranchii) diversification patterns through deep time. Biological Reviews 91, 950-981 (2016)."

Reference 51 "Froese R, Pauly D. FishBase World Wide Web electronic publication, Version

(01/2017). URL [Www Fishbase Org](http://www.fishbase.org) Accessed 1, (2017).”

Reference 52 “Team RDC. R: A language and environment for statistical computing). 3.4.2 edn. R Foundation for Statistical Computing (2017).” – I think this should be R Core Data Team or R Core Team.

Rob Cooke

REVIEWERS' COMMENTS

Reviewer #1 (Remarks to the Author):

I think the authors have done a sufficient job in addressing my feedback and comments in this updated manuscript. As such, I am now satisfied and recommend the paper be accepted.

Response:

We thank the reviewer for their comments, which allowed us to significantly improve our manuscript.

Reviewer #2 (Remarks to the Author):

Original comment:

Line 247. "FRic, FUn, FSp, and FUSE" Here the authors are not considering something important. Those metrics are scale dependent, here you are not comparing species from a defined ecological scale, such as Large Marine Ecosystems, Marine Ecoregions or other ecological classification. Instead, you are using the global scale as your gamma diversity and the grid cells as your alpha diversity. You are comparing a species from North Atlantic with a species from South Australia. This study would benefit by dividing the worlds oceans into specific ecological regions (biogeographical/environmental criteria) and then use each of those as your gamma diversity.

The suggestion here is:

- 1 - Divide the worlds ocean into ecoregions.
- 2 - Calculate the indices at species level and assemblage level for each of those ecoregions separately.

We thank the reviewer for bringing up this very important and valid issue. We see two situations that justify the use of regional species pools as references (gamma diversity) to calculate alpha (and beta) diversity metrics:

1. In the case where the aim of the study is to tease apart the relative influence of biotic interaction, dispersal limitation and environmental filtering in shaping local assemblage structure and composition (e.g., Suarez-Castro et al. 2022 and Mugnai et al. 2022), which is not the aim of our study.
2. In the case where the study aims at identifying biodiversity hotspots within regions (e.g., by dividing each SR, FRic and PD values at a grid-cell-scale by the species richness, functional volume and sum of branch lengths of the regional species pool, respectively), in order to point policy makers and environmental managers to specific localities within the region of interest for local conservation actions (e.g. Mouton et al. 2022). In that context, two cells with different absolute FRic values locally (e.g., one cell in the Central Indo-Pacific region and another in the Mediterranean Sea) may be considered as hotspots when standardizing by the (gamma) diversity of the region.

Our initial response:

Identifying biodiversity hotspots for each region independently is an interesting issue, but it goes beyond the scope of our study, which aims at highlighting spatial patterns of elasmobranch biodiversity at the global scale, specifically by (i) identifying regions of the world where elasmobranch biodiversity concentrates and (ii) determining whether these biodiversity hotspots are well covered by MPAs or overlap with human impacts (i.e., fishing) . Therefore, our work intends to help setting global conservation actions towards those regions where threatened elasmobranch biodiversity is found to be greatest.

Overall, we follow a classical approach in the field of conservation biogeography to identify global biodiversity hotspots (e.g., Orme et al. 2005, Albouy et al. 2017 and more recently Tietje et al. 2023) and we prefer to not follow a regional-based approach, which could be done in a future study with different objectives. It is also worth noting that regional-based studies are generally conducted using a finer spatial resolution (e.g., Mouton et al. 2022) and based on the compilation of fieldwork data, which therefore allows accounting for species abundances in the calculation of the different biodiversity metrics. We now discuss this point in the revised manuscript (L418-425: "While our global analysis represents an important step towards prioritizing conservation actions in the regions where threatened elasmobranch biodiversity is the greatest, it does not allow identifying biodiversity hotspots for each marine biogeographic realm³⁹. Future studies could complement our

findings by adopting a regional-based approach, where each region can be considered as independent pool of species. Coupled to finer distribution data at the regional scale (see Mouton et al.40) and to long-term abundance trends when available⁴¹, this could ultimately help policy and decision makers, and environmental managers, to set conservation actions towards threatened biodiversity at a regional scale (e.g., Mouillot et al.8).”

Importantly, our approach does not completely ignore scale-dependency given that our grid-cell analyses consider only co-occurring species (at the exception of FUN in the original manuscript, see below). Accordingly, in our biogeographic patterns section, we do not compare species from the North Atlantic with species from South Australia. Instead, we re-calculate FRic, FUN and FSp per grid cell (Fig. 2A-C). However, for FUN, we indeed originally calculated this index by considering the nearest neighbour within the global pool of species, and consequently accounted for species from different regions that do not necessarily co-occur (e.g., Violle et al. 2017). In light of the reviewer’s comment, in the revised manuscript, we have addressed this issue by modifying our calculation of FUN per grid so that instead of it being based on the nearest neighbour within the global pool of species, we are now using the nearest neighbour within co-occurring species. In that context, FUN reflects the degree to which assemblages are composed, on average, of functionally unique species at the local scale. In other words, this new approach considers the extent to which species co-occurring locally can play a similar functional role if another species were to disappear within this assemblage. We consider that it is also interesting to know, from a conservation perspective, whether assemblages are composed of functionally unique species at the global scale (i.e., composed of species that do not have close relatives in the world’s ocean). When considering this alternative approach, we found that FUN calculated based on co-occurring species or calculated considering all the species the world’s oceans, follow a similar spatial pattern (Figs. 2B and S10E), with FUN increasing towards the poles. This is an important result because as discussed by Violle et al. (2017), assemblages may be composed of functionally unique species locally but not regionally.

Reviewer answer:

I would like to thank the authors for providing further clarifications between lines L418-425 and for addressing the scale issue pointed in the previous review. The modification of the method applied, from using the nearest neighbour within the global pool of species to using the nearest neighbour within co-occurring species, provides a valid alternative to address the scale dependence issue.

Response:

We thank the reviewer for their feedback and are happy to learn that we have addressed their concerns.

Original comment:

Line 190. “Occupy almost the full extent (99%) of the functional space.” - The functional space shown at the Figure 1 A represents the functional space occupied by all species. That is = 100% of the species included in this study (100% = 1015 species). However, the functional space shown for the IUCN panel (bottom left) does not seem to display the same functional space structure as the other panels (constructed using all species as the other ones). Here red is the functional space occupied by threatened species (631 species) and green is the non-threatened (283 species). We then have 631 + 283 = 914 species, this means that this new functional space shown in this panel does not represent the 1015 species. Therefore, when the authors say that threatened species “occupy almost the full extent (99%) of the functional space” this new percentage given does not refers to the functional space for all species. In fact this new percentage refers to 914 species (1015 - 914 = 101 species less in caparison with the remaining panels). This may lead the reader to not accurate interpretation. Is there a reason why the authors are not using the same functional space structure in this IUCN panel as in the other ones?

Initial response:

We thank the reviewer for this comment, which made us realise there was a mistake in our code, whereby we accidentally excluded some species from this analysis. We have fixed this problem and have recalculated all IUCN Red List status-based indices based on the total number of species.

Although the values have slightly changed, the take-home message remains the same: threatened species occupy almost the full extent of the trait space: 98%. (L192-198).

Reviewer answer:

Thanks for addressing this point.

Response:

Thank you for helping us improve our manuscript.

Reviewer #3 (Remarks to the Author):

I previously reviewed this manuscript and believe the authors have done a good job at revising the manuscript. I really like the new and improved Fig. 1 and I think the additional trait imputation analyses give much more confidence to the reader about the robustness of the prioritization results. Overall, I believe the updated manuscript is much improved and I only have one remaining comment.

(References) The references probably need double-checking as there are a few weird issues/formatting errors.

For instance, reference 27 “union wc. IUCN Red list categories and criteria: version 3.1. IUCN Species survival commission.). IUCN Gland^ eU. KSwitzerlandCambridge UK (2001).”

Reference 28 “Guinot G, Cavin L. ‘Fish’(Actinopterygii and Elasmobranchii) diversification patterns through deep time. Biological Reviews 91, 950-981 (2016).”

Reference 51 “Froese R, Pauly D. FishBase World Wide Web electronic publication, Version (01/2017). URL [Www Fishbase Org](http://www.fishbase.org) Accessed 1, (2017).”

Reference 52 “Team RDC. R: A language and environment for statistical computing). 3.4.2 edn. R Foundation for Statistical Computing (2017).” – I think this should be R Core Data Team or R Core Team.

Rob Cooke

Response:

We thank the reviewer for his comments. We have checked our references and made sure there were no formatting errors. We believe perhaps the reviewer evaluated the tracked-changes version of our manuscript, which included the automatic EndNote references. However, we had also submitted a clean version where these issues do not appear. We have made sure that this submission only includes the clean version. Thanks for bringing this to our attention and in general, for helping us improve our manuscript.